# Specific glycine-dependent enzyme motion determines the potency of conformation selective inhibitors of threonyl-tRNA synthetase

Hang Qiao[1,7], Zilu Wang[1,7], Hao Yang [2,7], Mingyu Xia[1], Guang Yang[3], Fang Bai [2,4,5] ✉, Jing Wang [1,3] ✉ & Pengfei Fang [1,3,6] ✉

The function of proteins depends on their correct structure and proper dynamics. Understanding the dynamics of target proteins facilitates drug design and development. However, dynamic information is often hidden in the spatial structure of proteins. It is important but difficult to identify the specific residues that play a decisive role in protein dynamics. Here, we report that a critical glycine residue (Gly463) dominates the motion of threonyl-tRNA synthetase (ThrRS) and the sensitivity of the enzyme to antibiotics. Obafluorin (OB), a natural antibiotic, is a novel covalent inhibitor of ThrRS. The binding of OB induces a large conformational change in ThrRS. Through five crystal structures, biochemical and biophysical analyses, and computational simulations, we found that Gly463 plays an important role in the dynamics of ThrRS. Mutating this flexible residue into more rigid residues did not damage the enzyme's three-dimensional structure but significantly improved the thermal stability of the enzyme and suppressed its ability to change conformation. These mutations cause resistance of ThrRS to antibiotics that are conformationally selective, such as OB and borrelidin. This work not only elucidates the molecular mechanism of the self-resistance of OB-producing *Pseudomonas fluorescens* but also emphasizes the importance of backbone kinetics for aminoacyl-tRNA synthetase-targeting drug development.

Aminoacyl-tRNA synthetases (aaRSs) are key enzymes for protein translation in all cells, catalyzing the formation of an ester bond between a specific amino acid and the 3'-end adenosine of its conjugate tRNA[1,2]. There are three factors that make aaRSs valuable therapeutic targets. First, the sequence and fine structure differences between pathogenic microbial aaRS and human aaRS are applicable for designing drugs that selectively inhibit the pathogen aaRS[3,4]. Second, only low levels of aaRSs are required for protein translation under physiological conditions, and cells are able to withstand the down-regulation of aaRSs; therefore, drugs that specifically target hyperproliferative cells, such as

cancer cells, may be developed[5]. Third, some aaRSs have nonclassical functions associated with specific pathologies or cancers, making them potential targets for the development of novel pharmacological approaches[6–8].

AaRSs present multiple druggable pockets, including amino acid- and ATP-binding sites, tRNA-binding regions, and other binding sites, such as editing sites and auxiliary hydrophobic pockets[9]. Competitive binding to these sites with substrate molecules is a common mechanism of action (MoA) for aaRS inhibitors, including single-site[10–15], dual-site[16–20], and recently developed triple-site inhibitors[21]. In addition, inhibitors with novel

[1]State Key Laboratory of Chemical Biology, Shanghai Institute of Organic Chemistry, University of Chinese Academy of Sciences, 200032 Shanghai, China. [2]Shanghai Institute for Advanced Immunochemical Studies and School of Life Science and Technology, ShanghaiTech University, 393 Middle Huaxia Road, 201210 Shanghai, China. [3]School of Chemistry and Materials Science, Hangzhou Institute for Advanced Study, University of Chinese Academy of Sciences, 1 Sub-lane Xiangshan, 310024 Hangzhou, China. [4]School of Information Science and Technology, ShanghaiTech University, 393 Middle Huaxia Road, 201210 Shanghai, China. [5]Shanghai Clinical Research and Trial Center, 201210 Shanghai, China. [6]Guangdong Provincial Key Laboratory of Chiral Molecule and Drug Discovery, 510006 Guangzhou, China. [7]These authors contributed equally: Hang Qiao, Zilu Wang, Hao Yang. ✉e-mail: baifang@shanghaitech.edu.cn; JWang@sioc.ac.cn; FangPengfei@sioc.ac.cn

MoAs, including Trojan horses[22,23], induced-fit[24], and reaction hijacking inhibitors[25] have been reported. These inhibitors are highly valuable for understanding the catalytic mechanism of aaRS and for developing therapeutic drugs that target aaRS.

The natural product Obafluorin (OB) is a potent inhibitor of bacterial threonyl-tRNA synthetase (ThrRS)[26]. It is produced by *Pseudomonas fluorescens* ATCC 39502 through the nonribosomal peptide synthetase (NRPS) assembly line[26–32]. The NRPS ObiF contains a type I thioesterase (TE) domain with a rarely reported cysteine residue at the catalytic site that plays a critical role in the formation of the OB β-lactone ring[29]. OB has broad antibiotic activity against both Gram-positive and Gram-negative pathogens[26,27]. The catechol moiety of OB was found to be essential for its antibacterial activity[33]. Our previous work showed that OB covalently binds to ThrRS, forming an ester bond with a tyrosine residue in the catalytic center via the highly strained β-lactone ring[34], making OB the first covalent aaRS inhibitor with demonstrated crystal structures.

The elucidation of the self-resistance mechanisms of antibiotic-producing microorganisms will help to reveal the MoA of antibiotics, guide the discovery of new natural products, and provide key clues for clinical antibiotic resistance. The resistance of *P. fluorescens* to OB is related to the expression of ObaO, a second copy of ThrRS[26,33]. However, why ObaO is resistant to OB remains unknown.

During the analysis of the reason for the resistance of ObaO to OB, we found that a glycine residue involved in the formation of the active pocket of ThrRS contributes to the latter's sensitivity to OB. Through five crystal structures, biochemical and biophysical analyses, and computational simulations, we found that mutation of this flexible residue to a more rigid residue, such as a serine or an alanine, did not alter the structure, but significantly improved the thermal stability of the enzyme and suppressed the ability of ThrRS to change conformation. These mutations caused resistance of ThrRS to antibiotics that require conformational changes in ThrRS, including OB and another ThrRS inhibitor, borrelidin (BN). This work provides a deeper understanding of the MoA of OB and emphasizes the importance of backbone kinetics for drug development.

## Results

### Potential residues that cause OB resistance

ObaO, the second ThrRS in *P. fluorescens*, exhibits partial resistance to OB (Fig. 1a)[26]. Knockout of ObaO in *P. fluorescens* ATCC 39502 results in OB sensitivity, while expression of ObaO in OB-sensitive *Escherichia coli* strains confers resistance to OB[26]. However, the structure of ObaO predicted by Alphafold[35] is very similar to that of *E. coli* ThrRS (*Ec*ThrRS, Fig. 1b). In particular, the residues that interact with L-Thr, ATP, and tRNA in *Ec*ThrRS and ObaO are 100% conserved[36]. The reason why ObaO is resistant to OB is interesting. This immune protein is not only found in *P. fluorescens*, but also in *Chitiniphilus shinanonensis* and *Burkholderia diffusa*, which can produce OB in vivo[26] (Supplementary Fig. 1). In the *Ec*ThrRS–OB complex structure, 18 residues in the active center of ThrRS are located within 4.5 Å of the OB[34]. Among these residues, only Ala316 is substituted by an asparagine residue in the *P. fluorescens*, *C. shinanonensis*, and *B. diffusa* ObaO proteins. However, this substitution of alanine by asparagine also occurs in the *P. fluorescens*, *C. shinanonensis*, and *B. diffusa* housekeeping ThrRS proteins (Fig. 1c,

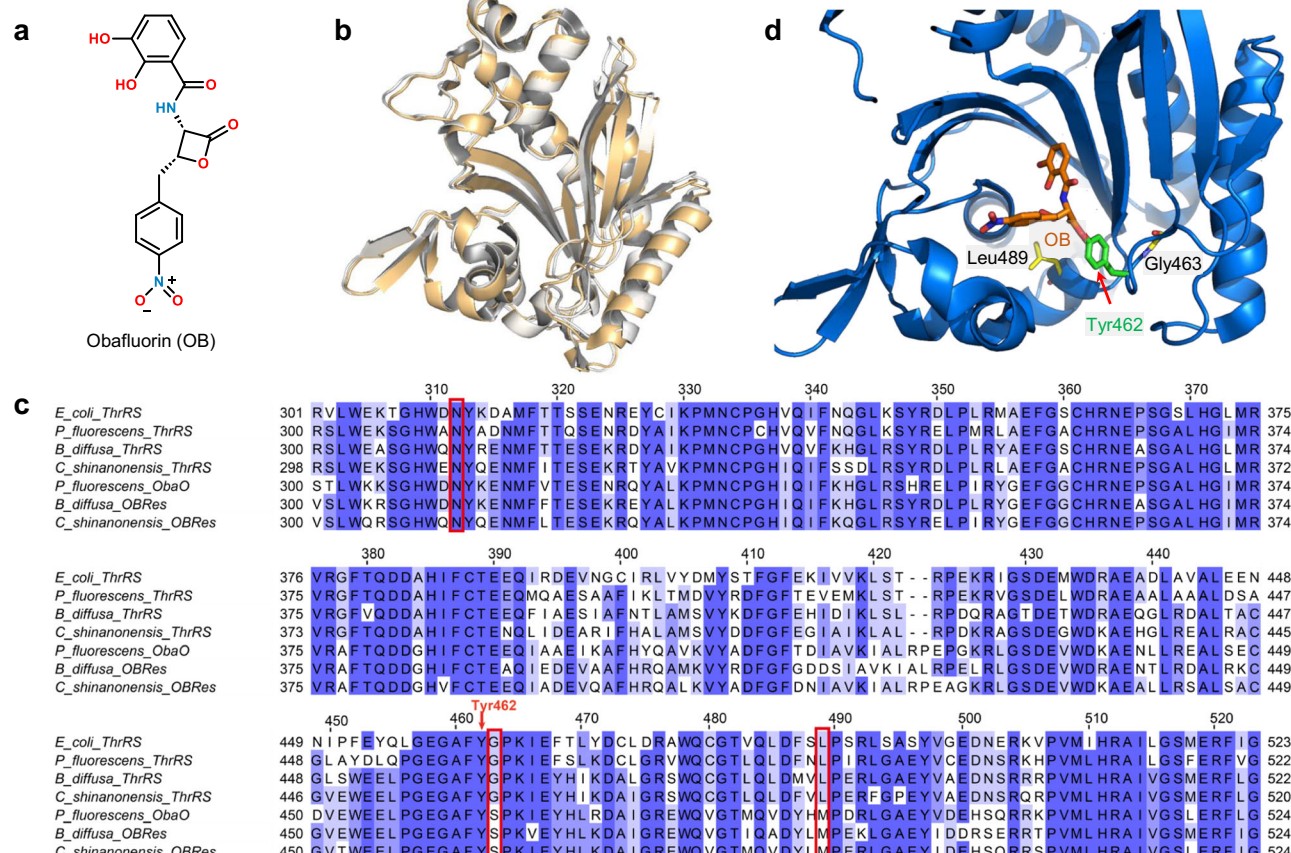

**Fig. 1 | Sequence alignment and structural analysis suggested that some residues may lead to OB resistance. a** Chemical structure of OB. **b** Superimposition of the structures of the catalytic domains of *Ec*ThrRS (gold, PDB code: 1EVK) and ObaO (white, predicted by AlphaFold). The RMSD is 0.740 Å over 280 Cα atoms. **c** Sequence alignment of ThrRS homologs. The alignment was performed by *Clustal* *Omega* and processed with *Jalview*. The alignment of Ala316, Gly463 and Leu489 corresponding to *Ec*ThrRS are highlighted in red boxes. **d** Zoomed-in view of the catalytic pocket (blue cartoon) of *Ec*ThrRS bound to OB (orange sticks). Tyr462 is shown as green sticks. Gly463 and Leu489 are shown as yellow sticks.

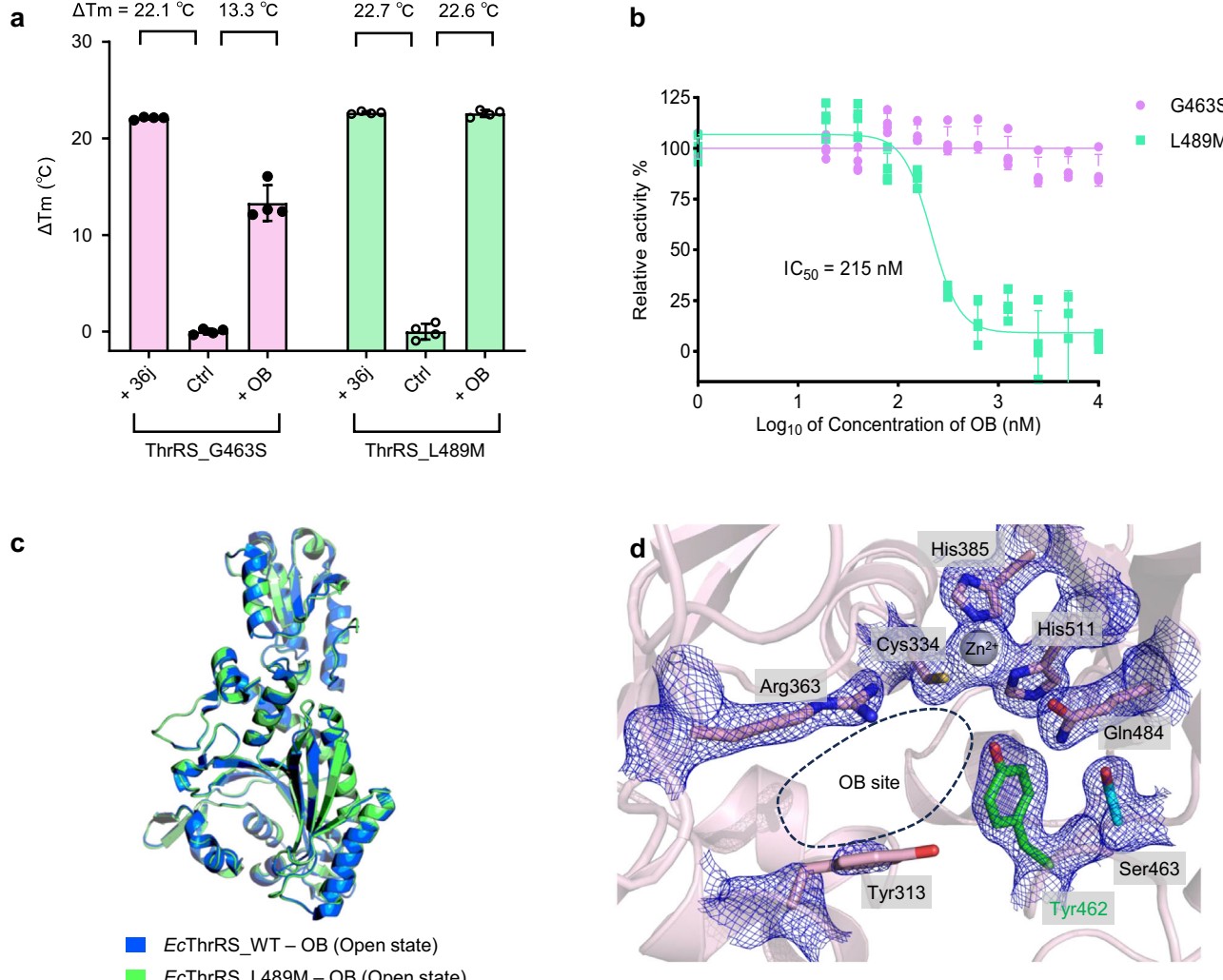

**Fig. 2 | Ser463 confers the resistance of OB to *Ec*ThrRS rather than Met489.**
**a** Diagram of the ΔTm values of *Ec*ThrRS_G463S and *Ec*ThrRS_L489M in the
presence or absence of OB and 36j. Error bars represent the standard deviations
($n = 4$, mean value ± SD). All the data points are shown as small circles.
**b** Inhibitory curves of OB on the ATP hydrolysis activity of *Ec*ThrRS_G463S or
*Ec*ThrRS_L489M. Error bars represent the standard deviations ($n = 4$, mean
value ± SD). All data points for EcThrRS_G463S and EcThrRS_L489M are shown
as pink and green dots, respectively. **c** Superimposition of the structures of

*Ec*ThrRS_WT–OB (marine cartoons) and *Ec*ThrRS_L489M–OB (green cartoons).
The RMSD is 0.160 Å over 385 Cα atoms. **d** Zoomed-in view of the catalytic pocket of
*Ec*ThrRS_G463S, which was crystallized in the presence of OB. The 2Fo-Fc electron
map (blue meshes contoured at 1.0 σ) is shown together with the structure
model. The Tyr462 residue and Ser463 residue are shown as green and cyan
sticks, respectively. The OB binding site is circled with black dashed lines. In
this state, the OB cannot form a covalent bond with Tyr462; hence, no density
for the OB can be seen. The density of Tyr313 is also poor in this state.

Supplementary Fig. 2). Using a thermal shift assay (TSA) and an ATP
hydrolysis assay[34], we tested whether mutating Ala316 to asparagine would
create OB resistance. Compound 36j, a substrate-competitive ThrRS inhi-
bitor, was used as an experimental control[21] (Supplementary Fig. 3). We
have shown that the influence of OB on the thermal stability of ThrRS (or
ThrRS mutants) is greater than or close to 36j when OB forms a covalent
bond with the enzyme, and much less than that of 36j when OB cannot form
a covalent bond with the enzyme[34]. The TSA results showed that OB
increased the mid-melting point (Tm) of *Ec*ThrRS_A316N by 33.4 °C,
which is slightly greater than that of 36j (Supplementary Fig. 4a). Con-
sistently, the ATP hydrolysis assay showed that OB had a strong inhibitory
effect on *Ec*ThrRS_A316N with an IC$_{50}$ value of 848 nM (Supplementary
Fig. 4b). Thus, the resistance of ObaO to OB is not due to the substitution of
residues directly interacting with OB.

We therefore looked for the replacement of residues that might
cause OB resistance in an expanded area in ObaO proteins. We noticed
that Gly463 and Leu489 are two strictly conserved residues in the ThrRS
proteins, but they are replaced by a serine and a methionine in the

ObaOs, respectively (Fig. 1c, Supplementary Fig. 2). In particular, Gly463
is adjacent to Tyr462, which is the residue covalently bound to OB, and
Leu489 is located within 7 Å of the OB in the pocket (Fig. 1d). We
postulated that the substitution of these two residues might prevent the
binding of ObaO to OB or hinder the formation of the covalent bond
between ObaO and OB.

## The glycine to serine substitution confers resistance of *Ec*ThrRS to OB

We expressed and purified the G463S and L489M *Ec*ThrRS mutants to
examine the effect of these two residue substitutions on OB resistance. We
utilized TSA to preliminarily estimate the binding affinities of
*Ec*ThrRS_G463S and *Ec*ThrRS_L489M to OB. The results showed that 36j
and OB increased the Tm of *Ec*ThrRS_L489M by 22.7 °C and 22.6 °C,
respectively (Fig. 2a), suggesting that OB strongly binds to
*Ec*ThrRS_L489M, similar to 36j. In contrast, 36j increased the Tm of
*Ec*ThrRS_G463S by 22.1 °C while OB increased the Tm of *Ec*ThrRS_G463S
by only 13.3 °C (Fig. 2a), suggesting that the binding affinity between OB

and $Ec$ThrRS_G463S is much weaker than that between 36j and $Ec$ThrRS_G463S. These results suggest that the G463S mutation weakens the binding affinity of ThrRS for OB, while the L489M mutation does not. Next, we studied the inhibitory activity of OB against $Ec$ThrRS_G463S and $Ec$ThrRS_L489M in an ATP hydrolysis assay. OB had a strong inhibitory effect on $Ec$ThrRS_L489M, with an $IC_{50}$ value of 215 nM (Fig. 2b). Therefore, the mutation of Leu489 to a methionine does not confer resistance of $Ec$ThrRS to OB. In contrast, OB showed poor activity against $Ec$ThrRS_G463S (Fig. 2b), with 5 μM OB inhibiting only approximately 10% of the activity of $Ec$ThrRS_G463S, indicating that mutating Gly463 to a serine renders $Ec$ThrRS resistant to OB.

We confirmed this finding by crystallization experiments (Supplementary Tables 1, 2). An $Ec$ThrRS_L489M–OB complex structure was successfully obtained by cocrystallizing 300 μM $Ec$ThrRS_L489M protein with 600 μM OB (Supplementary Fig. 5). The conformation of $Ec$ThrRS_L489M was the same as that of the WT protein (PDB: 8H98) when both were in complex with OB (Fig. 2c). Because in the OB-binding conformation, the active pocket has a larger opening[34], we refer to this state as the "open state". The MoA of OB in $Ec$ThrRS_L489M remained consistent with what we found in $Ec$ThrRS_WT[34]. In short, the β-lactone ring of OB opens and forms an ester bond with Tyr462; the $o$-diphenol group forms coordination bonds with the ThrRS conserved $Zn^{2+}$ ion; and the nitrobenzene group binds between Tyr313 and Arg363 (Supplementary Fig. 6). In contrast, cocrystallizing 300 μM $Ec$ThrRS_G463S protein with 600 μM OB did not result in an $Ec$ThrRS_G463S–OB complex structure. No electron density of OB was observed in the catalytic pocket of $Ec$ThrRS_G463S under this condition (Fig. 2d). The active pocket of this structure has neither an OB nor substrate molecules, so it is in an "apo state". The conformation in this state is similar to the "adenylation state" of the $Ec$ThrRS_Y462F–ATP complex structure we determined earlier[34], but slightly wider (Supplementary Fig. 7). This structure also indicates that although G463S weakens ThrRS binding to OB, it does not disrupt the active conformation of ThrRS.

To further confirm that this glycine is the key residue involved in OB sensitivity, we studied the corresponding reverse mutation (S464G) of the OB resistance gene ObaO. Similarly, we utilized a thermal shift assay and an ATP hydrolysis assay to test whether ObaO_S464G lost resistance to OB. The TSA results showed that OB increased the Tm of ObaO_WT by 3.2 °C but increased the Tm of ObaO_S464G by 14.1 °C (Supplementary Fig. 8a), suggesting that the S464G mutation can bind to OB. Consistently, the ATP hydrolysis assay showed that OB had no inhibitory effect on ObaO_WT but inhibited the activity of ObaO_S464G, with an $IC_{50}$ value of 3.9 μM (Supplementary Fig. 8b). Therefore, mutation of the corresponding serine to glycine restores the sensitivity of ObaO to OB.

Together, these results indicate that Gly463 is a key residue in ThrRS susceptibility to OB and that replacing it with a serine will induce OB resistance in ThrRS.

## The OB resistance of ThrRS mutant is not because of spatial repulsion

In the $Ec$ThrRS_G463S structure, the hydroxyl group of Ser463 is 5.0 Å away from the phenolic hydroxyl group of Tyr462 (Fig. 3a), which attacks the lactone structure of OB and forms a covalent bond with it[34]. Thus, the sidechain of Ser463 is too short to clash with the presumably bound OB and it is also unlikely that Ser463 will directly affect the nucleophilicity of Tyr462 and prevent it from forming a covalent bond with OB.

We then considered that Ser463 might have indirect effects through other OB-binding residues. In the apo state $Ec$ThrRS_G463S structure, the sidechain of the highly conserved residue Gln484 in the catalytic pocket forms a hydrogen bond with Tyr462, while in the OB-bound structure of the wild type (open state), the conformation of Gln484 is changed to accommodate OB (Fig. 3b). The distance between Ser463 and the conformation-altered Gln484 is only 1.6 Å (Fig. 3b). Therefore, if OB could

induce a conformational change in Gln484 of $Ec$ThrRS_G463S, there might be repulsion between the two residues. To test the possibility that Ser463 might become resistant to OB by blocking the conformational change of Gln484, we designed a G463S_Q484A mutant to remove the sidechain of Gln484 to avoid potential spatial conflicts. We expected that the G463S_Q484A mutant would recover sensitivity to OB. However, in the TSA experiment, OB increased the Tm of $Ec$ThrRS_G463S_Q484A by 17.2 °C, 10 °C less than the effect of 36j (Fig. 3c), suggesting that the $Ec$ThrRS_G463S_Q484A mutant, similar to $Ec$ThrRS_G463S, could still not be covalently inhibited by OB. Consistently, cocrystallizing 300 μM $Ec$ThrRS_G463S_Q484A protein with 600 μM OB did not yield an $Ec$ThrRS_G463S_Q484A–OB complex structure (Fig. 3d, Supplementary Table 3). The $Ec$ThrRS_G463S_Q484A structure is still in an apo state.

Therefore, G463S endows ThrRS with resistance to OB neither by altering the protein's structure nor by direct or indirect spatial repulsion.

## The glycine substitution restricts the conformational change of the active pocket of ThrRS

In the catalytic cycle of an enzyme, the conformation of its active pocket normally exhibits a series of dynamic states. At the same time, we observed that the OB binding conformation of ThrRS is distinctly different from the apo or adenylation conformation[34]. Since glycine is the only nonchiral residue with the greatest flexibility, we further proposed that the G463S mutation might induce resistance to OB by altering the molecular dynamics (MD) of the protein. To investigate the role of Gly463 in the movement of the enzyme, we performed all-atom MD simulations of $Ec$ThrRS_WT and the $Ec$ThrRS_G463S mutant. Arg363 and Ala460 are two residues located on opposite sides of the OB binding pocket of ThrRS. The distance between their centroids (denoted as $D_{R-A}$) can be used to characterize changes in the size of the pocket (Fig. 4a). In the MD simulation, the WT protein had two peaks in the frequency distribution curve, one corresponding to a $D_{R-A}$ of 13 Å and the other corresponding to a $D_{R-A}$ of 17 Å (Fig. 4b). In contrast, the $Ec$ThrRS_G463S mutant had only one peak at a $D_{R-A}$ of 14 Å (Fig. 4b). For reference, the $D_{R-A}$ is 17 Å when ThrRS binds OB and 12 Å when it binds ATP (Supplementary Fig. 9). This result suggests that the pocket of the WT protein has greater conformational variability and that the G463S replacement compresses its conformational distribution.

The dynamic characteristics of the pocket were also visualized using dynamic cross-correlation maps (DCCM)[37], a method that quantifies the correlation of movements between pairs of residues within a protein over the course of MD simulations. Specifically, positive correlation values observed in DCCM indicate synchronized movements between residue pairs, moving in the same direction[38], which usually reflects a structurally rigid interaction or coordination within the protein structure. Conversely, negative correlation values signify that the movements of residue pairs are opposite to each other, a phenomenon that often contributes to structural flexibility and dynamic interactions within the protein.

The N-terminal region of $Ec$ThrRS (residues 1–225) contains two domains involved in the proofreading of aminoacyl-tRNA (Supplementary Fig. 10). The constraints between this region and the catalytic domain (residues 242–535) or the anticodon-binding domain (residues 535–642) are weak. Therefore, the N-terminal region of $Ec$ThrRS was not included in our MD simulations. Residues 358–373 and residues 432–465 are from two opposite sides of the pocket (Supplementary Fig. 11). The motion of these two groups was negatively correlated in the MD trajectories of the WT protein, suggesting that the pocket of the WT protein is more dynamic (Fig. 4c, d). However, the motion of these two groups was positively correlated in the MD trajectories of the $Ec$ThrRS_G463S mutant, suggesting that the pocket of this mutant is relatively rigid (Fig. 4e, f).

These results suggest that the conformation of the active pocket of the WT protein is more flexible and can switch to a probable conformation to expose a larger binding pocket for the binding of OB, while the G463S mutation alters the conformational variability of the binding pocket of the protein and prevents the binding event (Supplementary Movie 1).

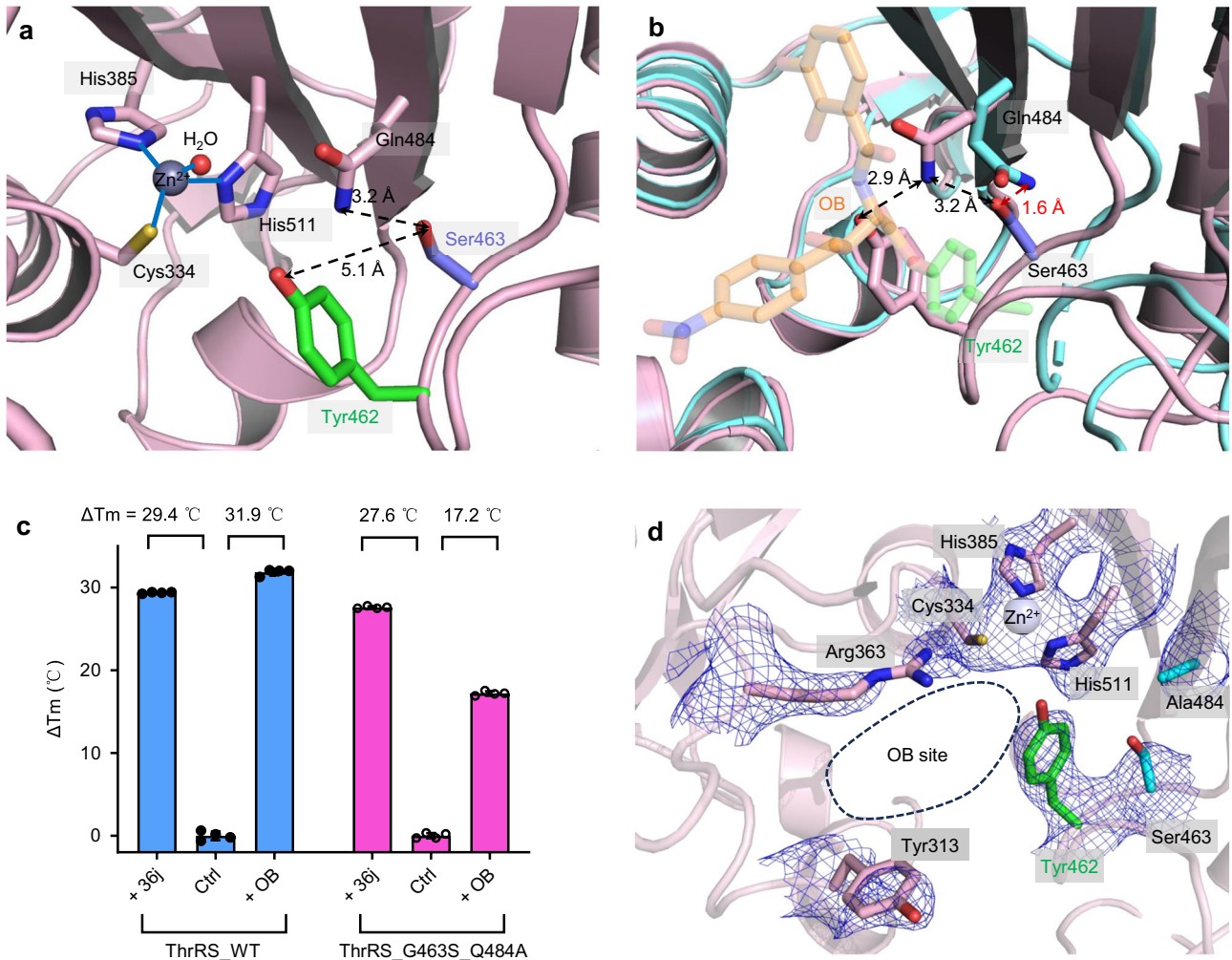

**Fig. 3 | Spatial interaction between Ser463 and Gln484 does not confer OB resistance. a** Zoomed-in view of the catalytic center of *Ec*ThrRS_G463S. Tyr462 is shown as green sticks, Ser463 is shown as slate sticks, and Gln484, Cys334, His385 and His511 are shown as pink sticks. **b** Superimposition of the structures of *Ec*ThrRS_G463S (pink cartoons) and *Ec*ThrRS_WT–OB (cyan cartoons). OB is shown as transparent orange sticks. **c** Diagram of the Tm values of *Ec*ThrRS_WT and *Ec*ThrRS_G463S_Q484A in the presence or absence of OB and 36j. Error bars represent the standard deviations ($n$ = 4, mean value ± SD). All the data points are shown as small circles. **d** Zoomed-in view of the catalytic pocket of *Ec*ThrRS_G463S_Q484A which was crystallized in the presence of OB. The 2Fo-Fc electron map (blue meshes contoured at 0.8 σ) is shown together with the structure model. Tyr462 is shown as green sticks. Ser463 and Ala484 are shown as cyan sticks. The OB binding site is circled with black dashed lines. In this state, the OB cannot form a covalent bond with Tyr462; hence, no density for OB can be seen.

## A small chiral residue substitution reproduces conformation limitation and OB resistance of ThrRS

Based on the above study, we speculated that G463S made ThrRS OB resistant due to the loss of the flexibility of the glycine residue, rather than due to the property of the serine hydroxyl sidechain. To test this speculation, we constructed an *Ec*ThrRS_G463A mutant. The TSA experiments showed that OB increased the Tm of *Ec*ThrRS_G463A by 19.5 °C, which was 14.0 °C less than that of the WT and 1.3 °C less than the Tm of *Ec*ThrRS_G463A increased by 36j (Fig. 5a). In addition, ATP hydrolysis experiments showed that OB had no inhibitory effect on *Ec*ThrRS_G463A (Supplementary Fig. 12), indicating that *Ec*ThrRS_G463A is also resistant to OB.

We also analyzed the conformational motion of *Ec*ThrRS_G463A by MD simulation and used $D_{R-A}$ to represent the size of the pocket. The frequency curve of *Ec*ThrRS_G463A had only one peak at a $D_{R-A}$ of approximately 14 Å, which was close to and slightly larger than that of *Ec*ThrRS_G463S (Supplementary Fig. 13). This finding suggests that the pocket of *Ec*ThrRS_G463A cannot be extended so that its $D_{R-A}$ reaches 17 Å and can be effectively suppressed by OB.

To further decipher the difference in conformational space between WT and mutant ThrRSs, we performed free energy landscape (FEL) analysis and obtained representative structures with the most stable conformations from the simulation ensembles. *Ec*ThrRS_WT shows a different profile of FEL from either G463S or G463A. Compared to the free energy landscapes of *Ec*ThrRS_G463S and *Ec*ThrRS_G463A, the energy basin of *Ec*ThrRS_WT covers a much larger range of radii of gyration (Rg), indicating that these conformations obtained along the trajectory experienced larger conformational changes (Fig. 5b-d). By comparing the conformation of residues 419-467 in the FEL nadir structure of the three proteins, we found that *Ec*ThrRS_WT had the most extended conformation, followed by G463A, and the narrowest was G463S (Fig. 5e). Furthermore, by comparing the FEL nadir conformation in the computational simulation and the real crystal structures, we found that the FEL nadir conformation of *Ec*ThrRS_WT was closer to the open state crystal structure, while the FEL nadir conformations of *Ec*ThrRS_G463S and *Ec*ThrRS_G463A were more similar to the apo state crystal structures (Supplementary Fig. 14). These analyses suggest that the G463S and G463A mutations affect the

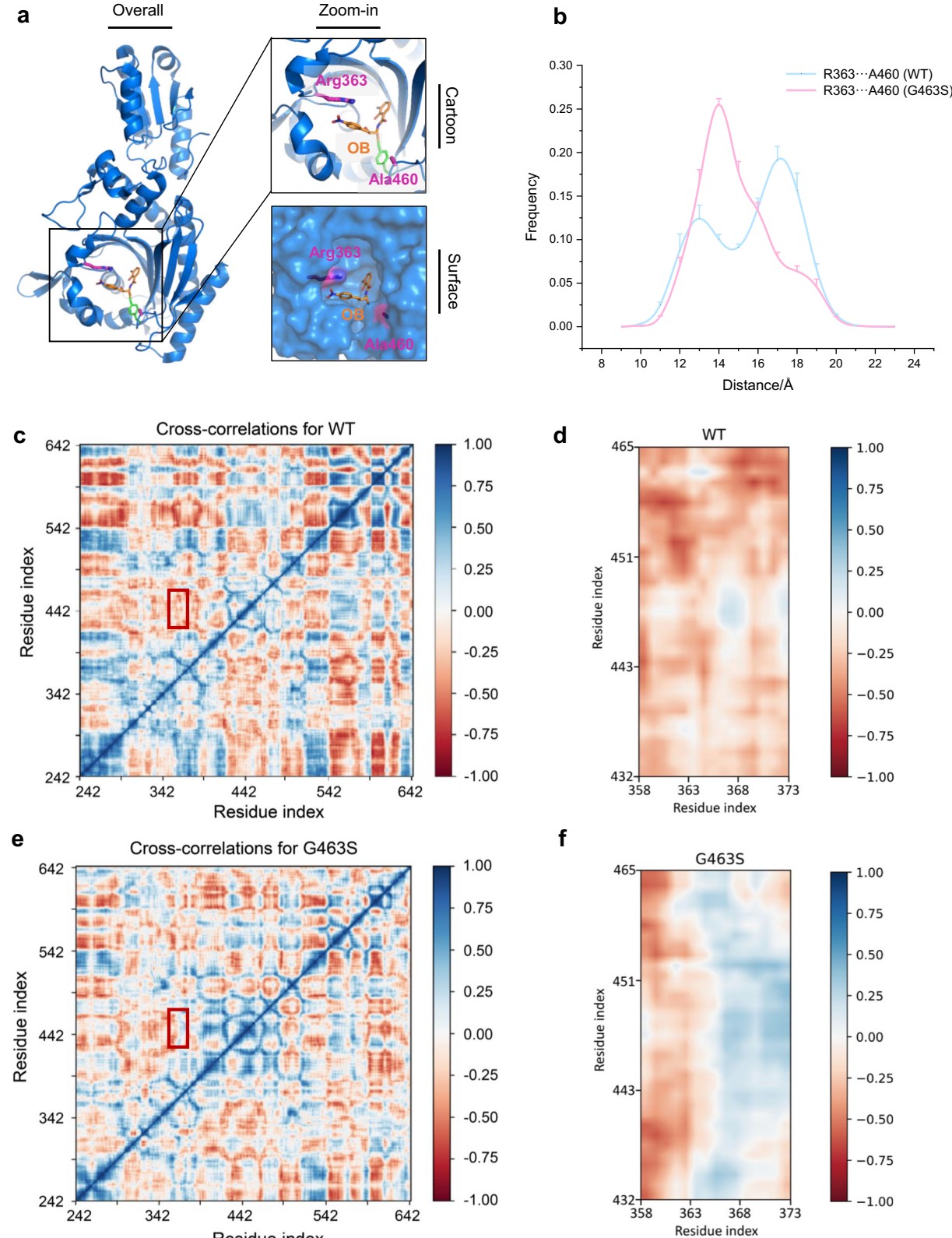

**Fig. 4 | The G463S mutation altered the dynamic properties of the active pocket.**
**a** Arg363 and Ala460 are located on both sides of the inlet of the catalytic pocket where OB binds. These two residues are shown as magenta sticks. **b** The G463S mutation alters the distribution of the conformational states of the protein *Ec*ThrRS. The distance between the centers of mass of the residues Arg363 and Ala460 is defined as the width of the catalytic pocket. The conformational space shifts of WT and G463S are shown as cyan and pink curves, respectively. Eight sets of simulations with different random initial velocities were performed and data were collected from 9 to 23 Å per angstrom (*n* = 8, mean value ± SD). **c, d** Dynamic cross-correlation maps (DCCM) of *Ec*ThrRS_WT. The dynamic cross-correlation matrix of Cα atoms around their mean positions is calculated. The extent of correlated motions and anticorrelated motions are color-coded from blue to red, which represents positive and negative correlations, respectively. (**d**) is a magnified view of the region in the red box in (**c**). **e, f** DCCM of *Ec*ThrRS_G463S. (**f**) is a magnified view of the region in the red box in (**e**).

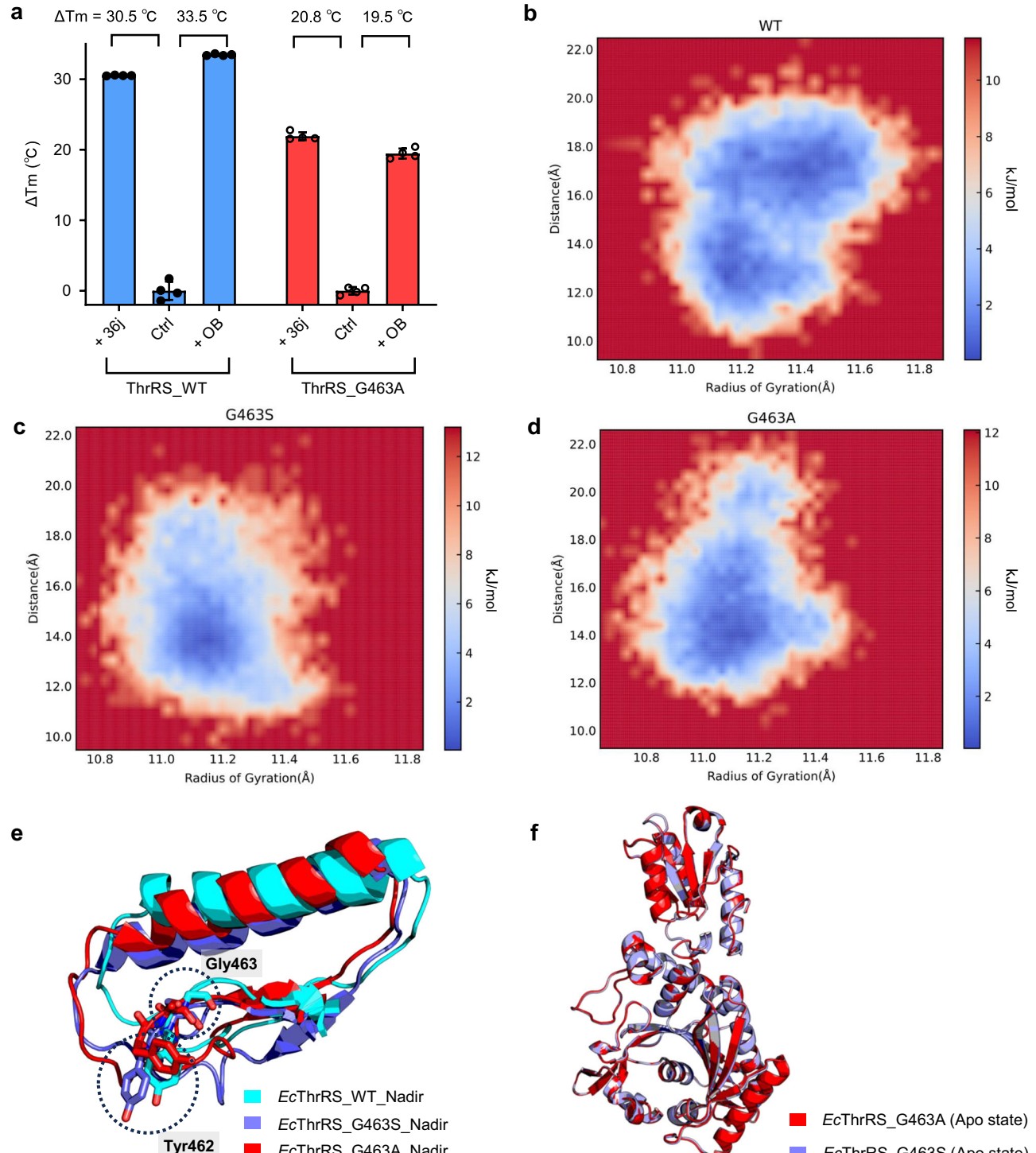

**Fig. 5 | The G463A mutation generates similar effects on the conformational change of ThrRS and blockage on the covalent binding of OB which were observed in the mutant G463S. a** Diagram of the $\Delta T_m$ values of EcThrRS_WT and EcThrRS_G463A in the presence or absence of OB and 36j. Error bars represent the standard deviations ($n = 4$, mean value ± SD). All the data points are shown as small circles. **b–d** Free energy contour maps derived from the radius of gyration (Rg) and RMSD values, where the dark blue area indicates a lower energetic conformation state. The Rg was calculated for residues 419–467. EcThrRS_WT exhibited a different free energy landscape (FEL) from those of both G463S and G463A, especially possessing a larger conformational tether of Rg. **e** Superimposition of the structure of residues 419-467 of EcThrRS_WT (cyan cartoons), G463S (slate cartoons) and G463A (red cartoons) at the FEL nadir corresponding to Fig. 5b–d. Residues at positions 462 and 463 are shown as sticks and circled with black dashed lines. **f** Superimposition of the structures of EcThrRS_G463A (red cartoons) and EcThrRS_G463S (slate cartoons). The RMSD is 0.343 Å over 318 Cα atoms.

conformational space of the ThrRS structure and cannot expose the active site for OB entry and reaction with Tyr462.

Similarly, we cocrystallized 300 μM EcThrRS_G463A with 600 μM OB and determined a 3.0 Å structure (Supplementary Table 4). As we predicted, the Tyr462 residue was not covalently modified by OB (Supplementary Fig. 15a) and the conformation of this structure was the same as that of EcThrRS_G463S (Fig. 5f). Most interestingly, there are some large but discontinuous electron densities near the ATP binding site as well as the zinc

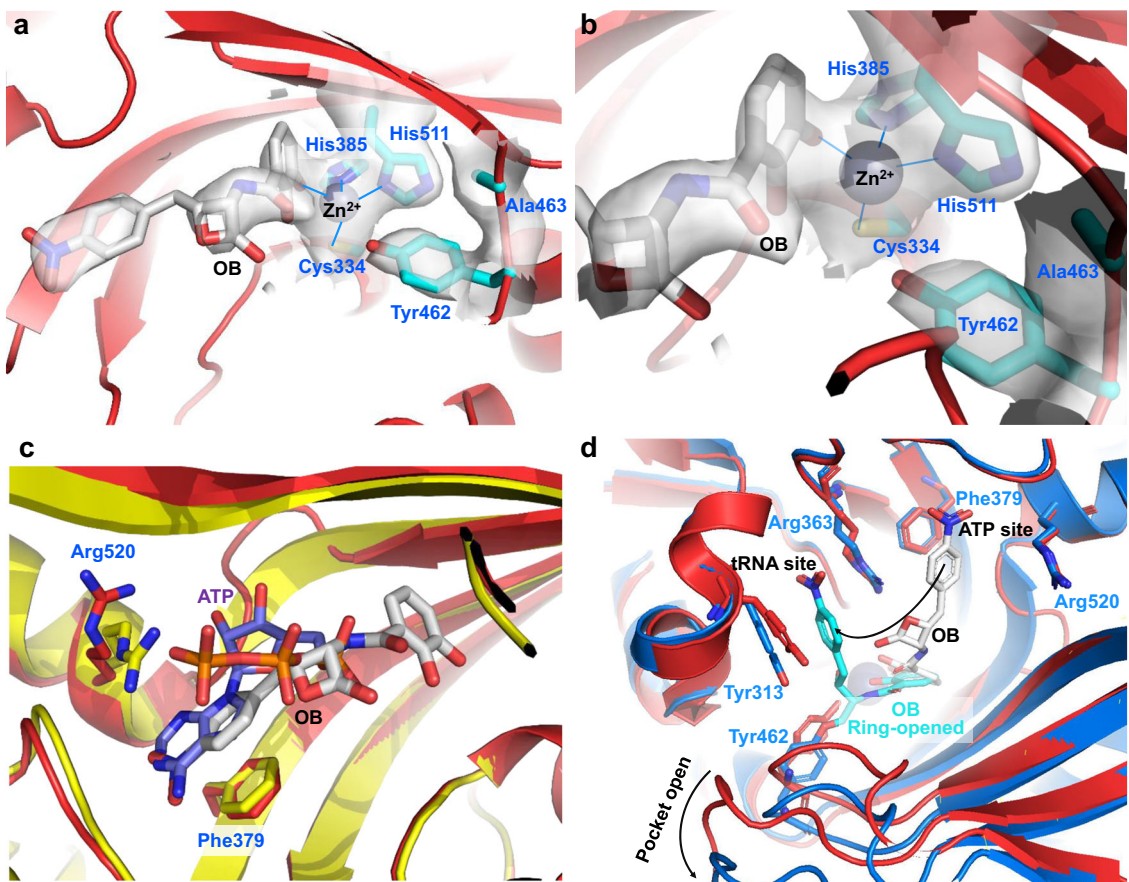

**Fig. 6 | OB noncovalently binds to *Ec*ThrRS_G463A without inducing a conformational change. a, b** Zoomed-in view of the catalytic pocket of *Ec*ThrRS_G463A, which was crystallized in the presence of OB. The 2Fo-Fc electron densities of OB, Cys334, His385, His511, Tyr462 and Ala463 (contoured at 1.0 σ) are shown as transparent surfaces. OB does not form an ester bond with Tyr462. **c** Superimposition of the *Ec*ThrRS_Y462F–ATP structure (yellow cartoons, PDB code: 8H99) with the *Ec*ThrRS_G463A–OB structure (red cartoons). The

nitrophenyl group of OB (in gray) binds between Phe379 and Arg520 where the adenine group of ATP (in purple blue) stacks. The orange parts in the middle of the panel are both terminal phosphate moieties of the cocrystallized ATP.
**d** Superimposition of the *Ec*ThrRS_WT–OB structure (marine cartoons, PDB code: 8H98) with the *Ec*ThrRS_G463A–OB structure (red cartoons). The ring-opened OB is shown as cyan sticks. The ring-closed OB is shown as gray sticks.

ion in this structure (Supplementary Fig. 15a). These electron densities can barely accommodate the OB structure (Fig. 6a, Supplementary Fig. 15b). We suspected that this is caused by unstable binding of OB in the ThrRS active pocket. In this case, one of the phenol hydroxyl groups forms a coordination bond with the zinc ion (Fig. 6b), and the nitrophenyl group inserts into the ATP binding site between Phe379 and Arg520 (Fig. 6c). This may be an intermediate state of OB binding in ThrRS. The conformation of the wild-type ThrRS can be further opened and the nitrobenzene portion of OB can instead bind to the tRNA binding site between Tyr313 and Arg363, thus bringing the lactone ring close to Tyr462 and reacting with it to form a new covalent bond (Fig. 6d, and Supplementary Movie 2). In contrast, the conformation of *Ec*ThrRS_G463A cannot be further opened, and in this binding mode, OB can be competed out by ATP (Supplementary Movie 3). Consistent with this assumption, when *Ec*ThrRS_G463A was cocrystallized in the presence of both OB and ATP, only ATP was resolved in the pocket (Supplementary Fig. 16, Supplementary Table 5). In summary, these results suggest that Gly463, which exploits the flexibility of a glycine residue, is a key site in determining the molecular dynamics of ThrRS.

### Conformation-constrained mutations cause resistance to conformation selective inhibitors

In addition to OB, another inhibitor molecule, borrelidin (BN, Fig. 7a), binds to ThrRS in an open conformation[39]. BN is an 18-member macrolide produced by *Streptomyces rochei* or *Streptomyces parvulus*. It binds ThrRS at a joint region of amino acid, ATP and tRNA pockets[39–41]. The

BN binding conformation is very similar to the OB binding conformation (Fig. 7b).

We have shown that the pocket conformation of the G463S and G463A mutants in the lowest energy state is too small for OB to stably bind. By the same principle, they may also have some resistance to BN. We confirmed this hypothesis by TSA and ATP hydrolysis experiments. BN increased the Tm of *Ec*ThrRS_G463S and *Ec*ThrRS_G463A by 17.4 °C and 19.1 °C, respectively, which were 16.5 °C and 14.8 °C lower than that of *Ec*ThrRS_WT (Fig. 7c). Similarly, BN had a strong inhibitory effect on *Ec*ThrRS_WT, with an IC₅₀ value of 559 nM in the presence of 2 μM ATP and 20 μM L-Thr, but the inhibitory effect of both *Ec*ThrRS_G463S and *Ec*ThrRS_G463A decreased by more than 20-fold (Fig. 7d). These results suggest that conformation-constrained mutations also render ThrRS resistant to other conformation-selective inhibitors, such as BN.

### Discussion
In summary, by studying the resistance of ObaO to OB, we found that a key glycine residue (Gly463) of ThrRS plays a dominant role in the molecular dynamics of the protein. Its substitution by small chiral amino acid residues significantly constrains the conformational space of ThrRS; therefore, we propose a general mechanism of ThrRS resistance to conformation-selective inhibitors, including OB and BN.

Gly463 is a highly conserved residue in the housekeeping ThrRSs. It is possible that the OB-resistant G463S substitution compromises enzyme activity. To our surprise, we found that the ATP hydrolysis activity of the

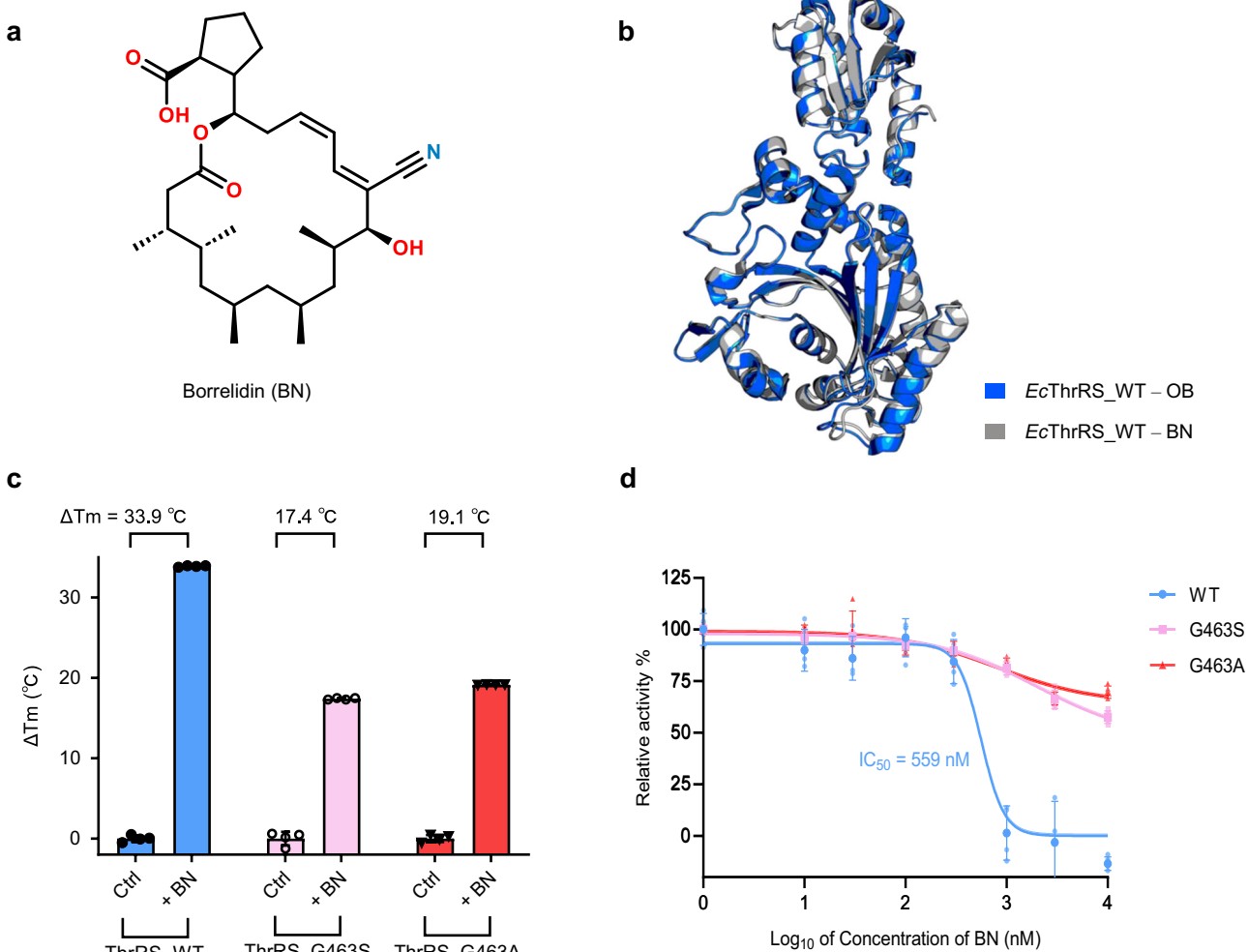

**Fig. 7 | Gly463 mutations also confer resistance of *Ec*ThrRS to BN.**
**a** Chemical structure of borrelidin (BN). **b** Superimposition of the structures of *Ec*ThrRS_WT–OB (PDB code: 8H98, marine cartoons) with *Ec*ThrRS_WT–BN (PDB code: 4P3P, gray cartoons). The RMSD is 0.396 Å over 345 Cα atoms. **c** Diagram of the ΔTm values of *Ec*ThrRS_WT/G463S/G463A in the presence or absence of BN. Error bars represent the standard deviations ($n = 4$, mean value ±

SD). All the data points are shown as small circles. **d** Inhibitory curves of BN on the ATP hydrolysis activity of *Ec*ThrRS_WT, G463S or G463A. Error bars represent the standard deviations ($n = 4$, mean value ± SD). All the data points for *Ec*ThrRS_WT/G463S/G463A are shown as pale blue dots, pale pink square dots, and pale red triangle dots.

Gly463 mutants was higher than that of the wild type (Supplementary Fig. 17). The mutation does not appear to have an adverse effect on the activity of the enzyme catalyzed amino acid activation. On the other hand, we noticed that the recombinant expression of Gly463 mutants produced more inclusion bodies than that of the wild type. Consistent with the discovery that Gly463 is critical to protein conformation, mutations in this residue may also sacrifice some of the cost of protein folding.

Functional proteins are in constant motion[42]. Structural flexibility allows proteins to adapt to their individual molecular binding partners and facilitates the binding process[43]. For example, the dynamics of bacterial leucyl-tRNA synthetase (LeuRS) allow for the whole catalytic cycle, while single mutations that affect global dynamics can reduce the catalytic efficiency of the enzyme[44]. Moreover, information such as the location and dynamic properties of drug binding pockets on proteins is critical for structure-based drug design and optimization of lead compounds[45].

Considering the dynamics of protein pockets during drug development may increase the accuracy of pocket identification and reveal alternative shapes or even a new transient pocket[43]. For example, a recent retrospective structure-based virtual screen (SBVS) study showed that the exploration of binding pocket dynamics in SBVS of sirtuin 2 inhibitors could achieve significant improvements in screening performance[46]; a combination of

experimental and computational methods of pocket dynamics successfully discovered an inhibitor that discriminates *Plasmodium falciparum* heat shock protein 90 from its human homolog[47].

Although not widely studied, MD is also thought to give insight into the reasons for potential resistance development. For example, a combination of bioinformatics analysis and MD simulations showed that the L528W mutation of Bruton's Tyrosine Kinase (BTK) reduces the conformational stability of BTK and decreased its binding affinity to ibrutinib compared to that of the WT, leading to drug resistance and potential disease recurrence[48].

In recent years, the development of drugs targeting aaRSs has received increasing amounts of attention. Our research suggested that molecular dynamics is an important factor in the design of drugs targeting aaRSs. This work not only elucidates the molecular mechanism of the self-resistance of OB-producing *P. fluorescens* that is equipped with a serine substitution at residue 464 (corresponding to Gly463 in *Ec*ThrRS) in the ObaO protein, but also emphasizes the importance of backbone kinetics, which is often hidden in static crystal structures, for aaRS-targeting drug development.

## Methods
### Protein purification
The *E. coli* ThrRS (residues 242-642, Supplementary Note 1) wild type, G463S, L489M, G463S_Q484A, G463A, A316N mutants, and the

*P. fluorescens* ObaO (residues 241-637) and S464G mutant were constructed in a pET28a vector attended with a 6×His-tag at the *C*-terminus. Each protein was expressed in the BL21 (*DE3*) strain and induced with 0.5 mM isopropyl-*β*-D-thiogalactoside for 20 h at 16 °C. The cell pellet (from 2 L) was lysed in buffer A (25 mM Tris pH 7.5, 500 mM NaCl, and 25 mM imidazole), loaded onto a Ni-Hitrap column (Cytiva, USA), washed with buffer A, and then eluted with a gradient increasing percentage of buffer B (25 mM Tris pH 7.5, 500 mM NaCl, and 250 mM imidazole) by a ÄKTA pure™ system (Cytiva, USA). The eluted protein was further purified by a Hitrap Q HP anion exchange column (Cytiva, USA) with gradient NaCl buffer (0.05–1 M NaCl in 25 mM Tris pH 7.5).

## Crystallization

For crystallization, the protein was further purified by a gel filtration S200 Increase column (Cytiva, USA) with a buffer containing 25 mM Tris pH 7.5, 200 mM NaCl, and 1 mM $MgCl_2$. All crystallization experiments were performed at 18 °C based on the sitting-drop method[49]. All proteins were concentrated to 15 mg/mL using a 10 kDa centrifugal filter (Millipore, USA). Before crystallization, OB (GlpBio, US) was mixed with protein at twice the molar ratio on ice for 1 h. ATP was mixed with protein at a fivefold molar ratio on ice for 1 h if added. For the microbatch crystallization screen, 0.5 μL of protein solution was mixed with an equal amount of precipitant solution (Molecular dimensions, UK) in microbatch 96-well plates using a Gryphon robot (ART technology, USA). Crystals grew to their final dimensions within 1–3 days.

The *Ec*ThrRS_G463S crystals were obtained from the condition of 0.02 M sodium formate, 0.02 M ammonium acetate, 0.02 M sodium citrate tribasic, 0.02 M potassium sodium tartrate tetrahydrate, 0.02 M sodium oxamate, 0.045 M imidazole, 0.055 M MES monohydrate pH 6.5, 20% v/v PEG 500 MME, and 10% w/v PEG 20000.

The *Ec*ThrRS_L489M–OB crystals were obtained from the condition of 2.0 M ammonium sulfate, and 0.15 M sodium citrate pH 5.5. 0.02 M DL-glutamic acid monohydrate, 0.02 M DL-alanine, 0.02 M glycine, 0.02 M DL-lysine monohydrochloride, 0.02 M DL-serine, 0.061 M Tris, 0.039 M bicine pH 8.5, 12.5% v/v MPD, 12.5% w/v PEG 1000, and 12.5% w/v PEG 3350.

The *Ec*ThrRS_G463S_Q484A crystals were obtained from the condition of 0.03 M magnesium chloride hexahydrate, 0.03 M calcium chloride dihydrate, 0.05 M sodium HEPES, 0.05 M MOPS pH 7.5, 20% v/v PEG 500 MME, and 10% w/v PEG 20000.

The *Ec*ThrRS_G463A–OB crystals were obtained from the condition of 0.02 M D-glucose, 0.02 M D-mannose, 0.02 M D-galactose, 0.02 M L-fucose, 0.02 M D-xylose, 0.02 M *N*-acetyl-D-glucosamine, 0.05 M sodium HEPES, 0.05 M MOPS pH 7.5, 12.5% v/v MPD, 12.5% w/v PEG 1000, and 12.5% w/v PEG 3350.

The *Ec*ThrRS_G463A–OB crystals were obtained from the condition of 0.03 M diethylene glycol, 0.03 M triethylene glycol, 0.03 M tetraethylene glycol, 0.03 M pentaethylene glycol, 0.05 M sodium HEPES, 0.05 M MOPS pH 7.5, 20% v/v ethylene glycol, and 10% w/v PEG 8000.

The *Ec*ThrRS_G463A–ATP crystals were obtained from the condition of 0.03 M magnesium chloride hexahydrate, 0.03 M calcium chloride dihydrate, 0.045 M imidazole, 0.055 M MES monohydrate pH 6.5, 20% v/v glycerol, and 10% w/v PEG 4000.

## Data collection and structure determination

The resulting crystals were flash-frozen in liquid nitrogen for data collection. The data were obtained from beamlines 10U2 or 02U1 at the Shanghai Synchrotron Radiation Facility (SSRF). Then, the datasets were indexed and integrated with XDS[50] or automatically processed by AutoPROC[51] at the Shanghai Synchrotron Radiation Facility (SSRF). The HKL files were scaled and merged with Aimless in the CCP4 suite[52]. The structures were determined by molecular replacement using the *Ec*ThrRS–OB structure (PDB: 8H98) or the *Ec*ThrRS structure (PDB: 1EVK) as a searching model in the Phaser program in the CCP4 suite[53]. After corrections for the bulk solvent and overall B values, the data were refined by iterative cycles of positional refinement and TLS refinement

with PHENIX[54] and model building with COOT[55]. All current models had good geometry and no residues were in the disallowed region of the Ramachandran plot. The data collection and model statistics are given in Supplementary Tables 1–5.

## Thermal shift assay

The *Ec*ThrRS_WT and L489M/G463S/G463S_Q484A/G463A/A316N mutant proteins, ObaO_WT and ObaO_S464G mutant proteins were prepared at 2 μM in buffer containing 25 mM Tris-HCl pH 7.5, 200 mM NaCl, and 20 μM OB (GlpBio, US) or BN (GlpBio, US) or $ddH_2O$ in equal volumes since the compounds were diluted with $ddH_2O$ to 1 mM as stocks. Compound 36j was assayed at the same final concentration as a positive control. SYPRO orange dye (Sigma, USA) was diluted in assay buffer containing 25 mM Tris-HCl pH 7.5 and 200 mM NaCl to a 40× concentration, and was added to the mixture to a final 4× concentration. Aliquots (20 μL) were added to a 96-well PCR plate. After complete mixing, the final solutions were heated from 25 to 95 °C at a rate of 0.015 °C/s, and fluorescence signals were monitored by QuantStudio 3 (Applied Biosystems by Thermo Fisher Scientific, USA). The melting temperature (Tm) was calculated through a previously established method[56,57].

## ATP hydrolysis assay

200 nM *Ec*ThrRS wide type and mutants, ObaO and ObaO_S464G mutant were incubated with serially diluted OB (0 to 10 μM for *Ec*ThrRS, 0 to 40 μM for ObaO) in a buffer containing 25 mM HEPES pH 7.5, 50 mM NaCl, 40 mM $MgCl_2$, 30 mM KCl, 0.01 mg/mL bovine serum albumin (BSA), and 0.004% Tween-20 at 37 °C for 4 h and then mixed 1:1 with a substrate mixture containing 4 μM ATP, 40 μM L-threonine, 25 mM HEPES pH 7.5, 50 mM NaCl, 40 mM $MgCl_2$, 30 mM KCl, 0.01 mg/mL BSA, and 0.004% Tween-20. The ATP hydrolysis reaction was performed at 37 °C for 4 h.

To evaluate the relative ATP hydrolysis rate of *Ec*ThrRS_WT, Y462F, G463S and G463A, a reaction mixture containing 200 nM *Ec*ThrRS, 2 μM ATP, 20 μM L-threonine, 25 mM HEPES pH 7.5, 50 mM NaCl, 40 mM $MgCl_2$, 30 mM KCl, 0.01 mg/mL BSA, and 0.004% Tween-20 was incubated at 37 °C for 4 h.

After incubation, the detection solution from the Kinase-Glo® luminescent kit (Promega, USA) was added to the reaction system at a 1:1 ratio, and the mixture was gently shaken for 10 min. The chemiluminescence signal was measured using a microplate reader (Tecan, USA). All experiments were performed in four replicates. The data were processed using GraphPad Prism 8 software.

## Molecular dynamics (MD) simulations

MD simulations were performed by using the Desmond[58] package of Schrödinger2023-1 with the OPLS4 force field[59]. The initial conformation of *Ec*ThrRS_G463S was obtained from the crystal structure determined in this work. To maintain consistency in the MD starting point and explore the inherent conformational space, the initial conformations of *Ec*ThrRS_WT and *Ec*ThrRS_G463A were obtained from a single mutation of the *Ec*ThrRS_G463S crystal structure. All systems were explicitly solvated with TIP3P water molecules[60] under cubic periodic boundary conditions for a 15 Å buffer region. The overlapping water molecules were removed, 0.15 M KCl was added, and the system was neutralized by adding $K^+$ as a counter ion. Brownian motion simulation was used to relax these systems into local energy minimum states separately. An ensemble (NPT) was then applied to maintain a constant temperature (310 K) and pressure (1.01325 bar) of the systems. For *Ec*ThrRS_WT and the two introduced mutants, 8 sets of simulations with different random initial velocities were started at 310 K, with each set lasting 400 ns. The simulations described above were performed using the AutoMD[61] (https://github.com/Wang-Lin-boop/AutoMD) script to handle the system and control the simulation process. To avoid artifacts caused by periodicity, we processed the trajectory using *trj_center.py*. The

distance between residue pairs was monitored by using *trajectory_asl_monitor.py*. The root mean square deviation (RMSD) and radius of gyration (Rg) were calculated by using *analyze_simulation.py*. Gromacs sham[62] and Matplotlib were used to calculate and visualize the three-dimensional Gibbs free energy surface with RMSD and Rg. The dynamic cross-correlation matrix (DCCM) was calculated and plotted by using the script *trj_essential_dynamics.py*. The process and analysis of trajectories described above were performed using the AutoTRJ (https://github.com/Wang-Lin-boop/AutoMD/blob/main/AutoTRJ) script. All of these simulation analysis scripts are from Schrödinger, Inc. The dynamics simulation movies were visualized by using VMD.

## Statistics and reproducibility
Enzymatic assay and thermal shift measurements were conducted in four repeats. MD simulations were conducted in eight repeats with different random initial velocities. Acquired data are presented as the mean values ± standard deviation (SD).

## Reporting summary
Further information on research design is available in the Nature Portfolio Reporting Summary linked to this article.

## Data availability
Atomic coordinates and structure factors for the reported crystal structures have been deposited with the Protein Data Bank under accession numbers 8WIH, 8WIG, 8WII, 8WIA, and 8WIJ. The source data behind the graphs are available as Supplementary Data 1.

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

## Acknowledgements

The authors thank Prof. Huihao Zhou for his helpful discussion and generous supply of compound 36j. We gratefully acknowledge the help from the staff of beamlines 10U2 and 02U1 at the Shanghai Synchrotron Radiation Facility. This work is supported by the National Key Research and Development Program of China grant 2022YFC2303100, National Natural Science Foundation of China grants 22277132 and 22277134, Shanghai Science and Technology Committee grant 22ZR1475000, the State Key Laboratory of Chemical Biology, Guangdong Provincial Key Laboratory of Construction Foundation, 2023B1212060022, and funding support from the National Key R&D Program of China (2022YFC3400500 and 2022YFC3400501) and the Shanghai Frontiers Science Center for Biomacromolecules and Precision Medicine at ShanghaiTech University. We also thank the HPC Platform of ShanghaiTech University.

## Author contributions

Investigation and methodology, H.Q., Z.W., H.Y., M.X. and G.Y.; review, editing, and writing, H.Q., Z.W., H.Y., F.B., J.W., and P.F.; conceptualization and supervision, F.B., J.W., and P.F. All authors read and approved the final manuscript.

## Competing interests

The authors declare no competing interests.
