## [Transparent Peer Review file · Communications Biology]

Specific glycine dependent enzyme motion determines the potency of conformation selective inhibitors of threonyl-tRNA synthetase

Corresponding Author: Professor Pengfei Fang

Figures originally included in the author's rebuttal have been redacted from this file.

Version 0:

Reviewer comments:

Reviewer #1

(Remarks to the Author)

The natural product antibiotic obafluorin (OB) represents the first covalent inhibitor of ThrRS, as well as of the whole aminoacyl-tRNA synthetase (aaRS) family, for which the inhibitory mechanism has been demonstrated through co-crystal structures. The OB-producing *P. fluorescens* itself employs a special ThrRS variant, named ObaO, to bypass the inhibition of bacterial ThrRS by OB. Understanding the basis of ObaO's resistance to OB may help to predict the potential resistance mutations in bacterial ThrRS and guide the development of new OB-based antibacterials. In this paper, Qiao et al. show that ObaO's resistance to OB is not due to simply gaining spatial conflicts or losing key interactions. Instead, the authors identified a critical glycine residue (Gly463) in the catalytic domain of OB-sensitive ThrRS, which is replaced by the more rigid residue serine in ObaO. The authors proposed that the single residue-determined backbone flexibility or rigidity confers the OB sensitivity to ThrRS or resistance to ObaO. This idea was validated by carefully-designed site-directed mutagenesis assays, biophysical and biochemical experiments, and molecular dynamics (MD) simulations. Importantly, this backbone flexibility-dependent resistance mechanism may also apply to other inhibitors, such as borrelidin. Overall, these findings from this study offer novel insights and may also aid in understanding the resistance mechanisms of other aaRS inhibitors. The data are robust, the conclusions are solid, and the manuscript is well organized. Below are some suggestions for improvements:

- 1) P3. The authors have published a paper for elucidating the inhibitory mechanism of OB previously, and they may have clearly introduced the background of the natural product OB there. However, for the convenience of the reader, I still recommend including more information about OB in this article. For example, is OB also active against eukaryotic ThrRS or is it specific for bacterial ThrRS? If the latter is the case, is there close evolutionary link between ObaO and eukaryotic ThrRS? This information may help to understand the origination of the resistance mechanism.
- 2) P3. "First, the sequence and topological difference between ..." The topology of catalytic domain should be quite conserved (at least more conserved than the 3D structure and the sequence) between microbial and human aaRS.
- 3) P4. "The resistance of *P. fluorescens* to OB is related to its expression of ObaO, a homolog of ThrRS." As my understanding, ObaO itself is a ThrRS with the same catalytic activity, and therefore, it may not be entirely accurate to say it is a "homolog" of ThrRS. ObaO is the second ThrRS in *P. fluorescens*? Or, *P. fluorescens* exclusively express ObaO and does not express a classical ThrRS.
- 4) P4. "..., while expression of ObaO in OB-sensitive *Escherichia coli* strains confers resistance to OB." Please provide a reference here.
- 5) P5. "However, this substitution of alanine by asparagine also occurs in the *P. fluorescens*, *C. shinanonensis*, and *B. diffusa* ThrRS proteins." Is there any evidence to show these ThrRS proteins are OB sensitive or not?
- 6) P5. "We noticed that Gly463 and Leu489 are two strictly conserved residues in the ThrRS proteins, ..." It is interesting why Gly463 and Leu489 are strictly conserved in ThrRS proteins. Compared to WT EcThrRS, is there any evidence to show that ObaO and G463/L489-mutated EcThrRS have sacrificed their catalytic activity or specificity to achieve OB resistance?
- 7) P8. "The dynamic characteristic of the pocket was also visualized using dynamic crosscorrelation maps (DCCM)." Please explain what positive and negative correlations on DCCM mean. Or, provide a reference here.
- 8) In Figure 4C and E, the protein sequence of ThrRS starts from 242. Why the N-terminal part of ThrRS was not analyzed?
- 9) Both RMSD and r.m.s.d are used as abbreviations for Root Mean Square Deviation in the manuscript. Need to be unified.

Reviewer #2

(Remarks to the Author)

This manuscript addresses several timely aspects of protein structure and dynamics and their relevance for antibiotic resistance. Threonyl-tRNA synthetase (ThrRS) is one of the twenty aminoacyl-tRNA synthetases (AARS) essential for translating the genetic code. The eukaryotic and prokaryotic versions of many, if not all AARS differ sufficiently that their two superfamilies are widely studied targets for antibiotic therapeutics. Within the bacterial clades, they are also of interest because bacteria are the source for many naturally occurring antibiotics, which are thought to reflect biochemical warfare among different bacterial strains. That creates a problem for strains that synthesize different antibiotics, because they need to develop resistant homolog genes in order to survive in the presence of the antibiotics they synthesize.

ThrRS is an example. The natural product molecule Obafluorin (OB) is a potent inhibitor of bacterial (ThrRS), and species that synthesize OB must develop ThrRS variants resistant to the compound. Previous work in the Fang laboratory have previously studied the structural biology and biochemical pharmacology of OB. In this manuscript they describe extensive studies of the structures of self-resistant (*P. fluorescens*) and sensitive (*E. coli*) ThrRS molecules that implicate the protein dynamics of the reaction in OB resistance. Specifically, they show that a glycine residue in the insertion domain between Motifs 2 and 3 of OB-sensitive *E. coli* ThrRS allows an off-path conformational change that promotes formation of a covalent bond between OB and an active-site tyrosine residue (Y462). Y462 enters the active site from what is often called the ID or insertion domain that likely represents a module acquired late in the evolution of Class II AARS and is located between Motifs 2 and 3. This domain undergoes a significant rotation toward the active site when OB is bound, leaving Motifs 2 and 3 more or less unchanged.

The authors use biophysical and biochemical analysis of various mutated residues to show the importance of the flexibility endowed by G463 in promoting the conformational change. That conformational change itself is necessary for forming a covalent bond to the OB. One key importance of these studies lies in the fact that G463 itself is outside the immediate environment provided for the binding surface of OB and thus must act cooperatively via conformational change. Moreover, thermal denaturation studies show that unless OB can form the covalent bond to Y662, its affinity is much reduced. These studies are fortified by a series of crystal structures that permit analysis of the structural consequences of the G462S and L489M mutations. The latter mutation does not materially change the affinity for OB; the former decreases the T_m by nearly two-fold. Finally, enzymatic assays confirm that ATP-dependent threonine activation is almost the same with the L489M mutant and with the G463S mutant. Thus, the G463S mutant, which prevents the conformational change, also confers OB resistance.

Altogether, this paper was a pleasure to read. It describes a robust structural and mechanistic enzymological analysis of OB resistance and localizes it to a single residue that also slows or limits the off-path conformational change that leads to strong inhibition. It thus merits publication in *Nature Communications*.

There are numerous infelicitous examples of poor English usage that should be addressed in a minor revision before publication. One example is in the caption to Figure 6, in the sentence: "es. OB not forms an ester bond with Tyr462". I presume the authors intended to write something along the lines of es. "OB does not form an ester bond with Tyr462". The authors should consult an English speaker for assistance with the entire manuscript.

Charles W. Carter, Jr

Reviewer #3

(Remarks to the Author)

Pengfei Fang et al. discuss in their manuscript how the potency of active site selective inhibitors of threonyl-tRNA synthetase is determined by the dynamics of the protein which in turn is determined by a specific glycine in the active site. This critical glycine residue (Gly463) allows the dynamic motion of threonyl-tRNA synthetase (ThrRS) either to bring binders in the correct position to allow covalent reaction (e.g. obafluorin), or to increase the interaction with the ligand so as to firmly occlude the active site as with borrelidin. Mutation of this glycine to a steric more demanding residue afforded a more rigid protein with higher thermal stability, and thus suppressed its ability to change conformation. As a result these mutations caused resistance of ThrRS to antibiotics that are conformationally selective.

The dynamics of the active site as described here, is a bit reminiscent of the in this journal recently described conformational changes as noticed for the class Ia LeuRS (doi.org/10.1038/s42003-022-03825-8). The dynamics as highlighted in this manuscript allowed for the whole catalytic cycle, and likewise single mutants were described influencing the overall dynamics and reducing the catalytic efficiency for this particular enzyme.

In the present manuscript the requirement for the described dynamics to enable the antibacterial effects of the studied antibiotics has been clearly shown. It is unclear however, how the catalytic efficiency is influenced by the proposed mutants. This brings us to conclude some additional assays are warranted to clearly study the effects of the proposed mutations on

the enzymatic activity.

The additional movies are an excellent addition to illustrate the manuscript findings.

Overall, this is a very interesting paper generally well formulated, but we have following remarks and questions on the proposed text, requiring some major but feasible changes.

1. According to the authors, the mutants do not disrupt the active conformation of ThrRS. However, do these mutants affect the catalytic activity of the enzyme? No data are provided.
If there is also no negative effect found on the enzymatic activity, the change of binding affinity of OB by the ATP hydrolysis assay is more convincing.
2. It is better to give some definition or name for each conformational states as otherwise it is quite difficult for the reader to understand what you want to explain in the manuscript.
3. If the serine residue in ObaO refers to the resistance for OB, does mutation of this residue to glycine restore the sensitivity of ObaO to OB?
4. In this manuscript, 36j was selected as the positive control to compare with OB when examining the potential binding affinity. However, by checking the reference paper for structure 36j (<https://doi.org/10.1021/acs.jmedchem.2c00134>), the structure shown in Figure S2 is actually compound 36k in the cited paper. This is rather confusing to the reader. Compound 36k actually shows the best antibacterial activity, while 36j has the higher affinity but presumably less uptake. Please correct the structure.
5. For TSA measurements (Fig 2,3,5), the results for 36j are also a bit strange. For these four different proteins including ThrRS WT, ThrRS_G436A ThrRS_G436S, ThrRS_G436S_Q484A, the reported ΔT_m is 29.4, 20.8, 22.1, 27.6, which in our opinion does not make sense according to the proposed theory. In this manuscript, G436S should provide the worst result, as the serine sidechain is bulkier and provides more constraint and the overall structure should be the least flexible. But the ΔT_m value of ThrRS_G436A is lower than for ThrRS_G436S? In addition, the double mutation has better binding affinity for 36j and OB (ΔT_m increased comparing the double mutant with G436S); how to explain these data?
If the TSA measurements cannot fully explain the proposed theory, the real binding affinity should be measured by a biochemical or biophysical assay (preferable action).
6. Figure 2D: the electron map shown does not really correspond with the residues? Suggestion to the authors: please reconstruct the density map, the selected residues can be shown with the electron density while leaving other residues shown in sticks. This will help clarify the content.
7. Figure 3 – the title: “Interaction of introduced Ser463 and conserved Gln484 doesn’t prevent the covalent binding of OB” is confusing. We believe the presence of Ser does prevent covalent binding, (no reaction with Tyr462) but not the less deeper binding of OB. What the authors most probably mean is that the interaction between Ser463 and Gln484 is not the reason for lack of covalent binding of OB. The title can be changed to “Spatial interaction between Ser463 and Gln484 does not confer OB resistance”.
8. Figure 3 Panel D: once more, electron map and structure model do not seem to correspond?? We suggest to redo the figure as for Figure 2D and only show the density of selected residues. In addition, in this state, the OB cannot form a covalent bond with Tyr462, and hence no density for OB can be seen, but for clarity such statement should be added here to avoid confusion.
9. Figure 6: Phe378 in figure 6C needs to be corrected to Phe379. In addition, the stacking of nitrobenzyl between Arg520 and Phe379 can be (difficultly) seen, this group is superposed with adenine in this view. Hence, preferably add to the text “The nitrophenyl group of OB (in grey) binds ...”. For clarity, also include in text that the orange parts are both terminal phosphate moieties of the co-crystallized ATP.
10. Please use the required format to provide all references. (e.g. the title format of references is not consistent).

Additional detailed remarks:

Page 6 line 3: please add the reference for the assay used.

Page 7 line 10: to clarify, add: while in the OB-bound structure “of the wild type”

Page 9 halfway: we believe this can be better phrased as “so that it can be suppressed by OB when D_{R-A} reaches 17 Å.”

Page 9 penultimate line: “nadirs”?

Page 10 line 2: “WT is closer to the OB-bound form”. We are a bit lost here. Is the WT structure not the same as for the apo protein? Or did the authors mean the WT in presence of ATP?

Page 12 line 5: “MD is also thought to influence drug resistance”. This seems not correctly formulated; MD itself not influences drug resistance, but rather gives insight into the reasons for potential resistance development.

Page 12 penultimate line: “OB-producing *P. fluorescens*”. Maybe add in parenthesis for clarity: “(equipped with the G463S mutation in the ObaO protein)”.

Page 13: the key resources table is obviously important but it is rather awkward to have a 3 page table in the main section. This could alternatively be in the supplementary section. I leave this remark to the discretion of the editor.

Page 15 line 5 from bottom: typo; correct to "washed".

Figure 1D: for convenience, use an arrow to pinpoint either the Tyr462 or Gly463 residue.

Figure 5: correct the second part of the title to "blockage on the covalent binding of OB which were observed in the mutant G463S."

Figure 5 panel E: please indicate some residues on the cartoon, at least position 463.

Figure 7 title: should read "resistance of EcThrRS to BN"

Table S2: please add the PDB code in the table.

Author Rebuttal letter:

RE: manuscript COMMSBIO-24-1169 by Qiao et al.

All reviewers provided thoughtful and helpful comments on our work. We considered and discussed these comments very carefully. Listed below are point-by-point responses to each comment of each reviewer.

Reviewer #1 (Remarks to the Author):

The natural product antibiotic obafluorin (OB) represents the first covalent inhibitor of ThrRS, as well as of the whole aminoacyl-tRNA synthetase (aaRS) family, for which the inhibitory mechanism has been demonstrated through co-crystal structures. The OB-producing *P. fluorescens* itself employs a special ThrRS variant, named ObaO, to bypass the inhibition of bacterial ThrRS by OB. Understanding the basis of ObaO's resistance to OB may help to predict the potential resistance mutations in bacterial ThrRS and guide the development of new OB-based antibacterials. In this paper, Qiao et al. show that ObaO's resistance to OB is not due to simply gaining spatial conflicts or losing key interactions. Instead, the authors identified a critical glycine residue (Gly463) in the catalytic domain of OB-sensitive ThrRS, which is replaced by the more rigid residue serine in ObaO. The authors proposed that the single residue-determined backbone flexibility or rigidity confers the OB sensitivity to ThrRS or resistance to ObaO. This idea was validated by carefully-designed site-directed mutagenesis assays, biophysical and biochemical experiments, and molecular dynamics (MD) simulations. Importantly, this backbone flexibility-dependent resistance mechanism may also apply to other inhibitors, such as borrelidin. Overall, these findings from this study offer novel insights and may also aid in understanding the resistance mechanisms of other aaRS inhibitors. The data are robust, the conclusions are solid, and the manuscript is well organized.

Response: We are very grateful to the reviewer for the understanding of the significance of our research and the general approval of this work.

Below are some suggestions for improvements:

1) P3. The authors have published a paper for elucidating the inhibitory mechanism of OB previously, and they may have clearly introduced the background of the natural product OB there. However, for the convenience of the reader, I still recommend including more information about OB in this article. For example, is OB also active against eukaryotic ThrRS or is it specific for bacterial ThrRS? If the latter is the case, is there close evolutionary link between ObaO and eukaryotic ThrRS? This information may help to understand the origination of the resistance mechanism.

Response: We appreciate the reviewer's suggestion. We have updated the background of the natural product OB in the revised manuscript Page 3 line 22-Page 4 line 3. We also attach it below for the reviewer's convenience.

The natural product Obafluorin (OB, Fig. 1A) is a potent inhibitor of bacterial threonyl-tRNA synthetase (ThrRS)²⁶. It is produced by *Pseudomonas fluorescens* ATCC 39502 through the non-ribosomal peptide synthetase (NRPS) assembly line^{26&32}. The NRPS ObaF contains a type I thioesterase (TE) domain with a rarely reported cysteine residue at the catalytic site that plays a critical role in the formation of the OB β -lactone ring²⁹. OB has a broad antibiotic activity against both Gram-positive and Gram-negative pathogens^{26,27}. The catechol moiety of OB was found to be essential for its antibacterial activity³³. Our previous work showed that OB covalently binds to ThrRS, forming an ester bond with a tyrosine residue in the catalytic center via the highly strained β -lactone ring³⁴, making OB the first covalent aaRS inhibitor with demonstrated crystal structures.

References:

26 Scott, T. A. et al. Immunity-Guided Identification of Threonyl-tRNA Synthetase as the Molecular Target of Obafluorin, a beta-Lactone Antibiotic. *ACS chemical biology* 14, 2663-2671 (2019).

27 Wells, J. S., Trejo, W. H., Principe, P. A. & Sykes, R. B. Obafluorin, a novel beta-

lactone produced by *Pseudomonas fluorescens*. Taxonomy, fermentation and biological properties. *The Journal of antibiotics* 37, 802-803 (1984).

28 Kumar, P. et al. L-Threonine Transaldolase Activity Is Enabled by a Persistent Catalytic Intermediate. *ACS chemical biology* 16, 86-95 (2021).

29 Schaffer, J. E., Reck, M. R., Prasad, N. K. & Wencewicz, T. A. beta-Lactone formation during product release from a nonribosomal peptide synthetase. *Nat Chem Biol* 13, 737-744 (2017).

30 Scott, T. A., Heine, D., Qin, Z. & Wilkinson, B. An L-threonine transaldolase is required for L-threo-beta-hydroxy-alpha-amino acid assembly during obafuorin biosynthesis. *Nature communications* 8, 15935 (2017).

31 Kreitler, D. F., Gemmell, E. M., Schaffer, J. E., Wencewicz, T. A. & Gulick, A. M. The structural basis of N-acyl-alpha-amino-beta-lactone formation catalyzed by a nonribosomal peptide synthetase. *Nature communications* 10, 3432 (2019).

32 Jones, M. A. et al. Discovery of L-threonine transaldolases for enhanced biosynthesis of beta-hydroxylated amino acids. *Communications biology* 6, 929 (2023).

33 Batey, S. F. D. et al. The catechol moiety of obafuorin is essential for antibacterial activity. *RSC chemical biology* 4, 926-941 (2023).

34 Qiao, H. et al. Tyrosine-targeted covalent inhibition of a tRNA synthetase aided by zinc ion. *Communications biology* 6, 107 (2023).

Whether OB has inhibitory activity against eukaryotic ThrRS is also a great question. Because the key glycine residue associated with OB sensitivity is generally conserved in eukaryotic ThrRS, we hypothesize that OB also has a certain degree of inhibitory activity against eukaryotic ThrRS. We performed an in vitro translation assay based on the extraction of *Kluyveromyces marxianus*, which is a eukaryotic microorganism. The specific glycine in *K. marxianus* ThrRS (Uniprot ID: W0TDP3) is Gly556. OB inhibited the activity of this system with an IC₅₀ value of about 9 μM (We attach the data below for the reference of the reviewer.). Consistent with the literature by Scott, T. A. et al. (*ACS Chem. Biol.* 2019, 14, 2663-2671), there is no specific evolutionary link between ObaO and eukaryotic ThrRS.

We have included a maximum likelihood tree of ThrRS in the new Supplementary Fig. 1.

2) P3. First, the sequence and topological difference between ... The topology of catalytic domain should be quite conserved (at least more conserved than the 3D structure and the sequence) between microbial and human aaRS.

Response: We agree with the reviewer and have revised it as the sequence and fine structure differences. (Page 3 line 5)

3) P4. The resistance of *P. fluorescens* to OB is related to its expression of ObaO, a homolog of ThrRS. As my understanding, ObaO itself is a ThrRS with the same catalytic activity, and therefore, it may not be entirely accurate to say it is a homolog of ThrRS. ObaO is the second ThrRS in *P. fluorescens*? Or, *P. fluorescens* exclusively express ObaO and does not express a classical ThrRS.

Response: We have adopted the reviewer's suggestion to replace a homolog of ThrRS with a second copy of ThrRS (Page 4 line 7) or the second ThrRS (Page 4 line 23).

4) P4. ..., while expression of ObaO in OB-sensitive *Escherichia coli* strains confers resistance to OB. Please provide a reference here.

Response: We have added the citation of the reference Scott, T. A. et al. Immunity-Guided Identification of Threonyl-tRNA Synthetase as the Molecular Target of Obafuorin, a β-Lactone Antibiotic. *ACS Chem. Biol.* 14, 2663-2671 (2019). (Page 4 line 25).

5) P5. However, this substitution of alanine by asparagine also occurs in the *P. fluorescens*, *C. shinanonensis*, and *B. diffusa* ThrRS proteins. Is there any evidence to show these ThrRS proteins are OB sensitive or not?

Response: We thank the reviewer for this constructive question. There is no direct biochemical evidence to show these ThrRSs proteins are OB sensitive or not. In order to more directly rule out the possibility that this substitution of alanine by asparagine causes resistance, we performed new thermal shift assay (TSA) and ATP hydrolysis assay. The TSA result showed that OB increased the T_m value of EcThrRS_A316N by 33.4 °C, which is slightly higher than 36j (new Supplementary Fig. 4A). Consistently, the ATP hydrolysis assay showed that OB had a strong inhibitory effect on EcThrRS_A316N with an IC₅₀ value of 848 nM (new Supplementary Fig. 4B). Therefore, A316N is not a mutation that causes OB resistance.

We have added these results to Page 5 line 15-18. The new Supplementary Fig. 4 is shown below.

6) P5. We noticed that Gly463 and Leu489 are two strictly conserved residues in the ThrRS proteins, ... It is interesting why Gly463 and Leu489 are strictly conserved in ThrRS proteins. Compared to WT EcThrRS, is there any evidence to show that ObaO and G463/L489-mutated EcThrRS have sacrificed their catalytic activity or specificity to achieve OB resistance?

Response: The reviewer raised a great point. Gly463 is a highly conserved residue in the housekeeping ThrRSs. We have also suspected that the G463S mutation that produces OB resistance compromises enzyme activity. To our surprise, we found that the ATP hydrolysis activity of the Gly463 mutants was higher than that of the wild type (new Supplementary Fig. 17). The mutation does not appear to have an adverse effect on the activity of the enzyme catalyzed amino acid activation. We admit that the interference with the transfer of amino acids from aminoacyl-adenylate to tRNA cannot be ruled out yet. On the other hand, we found that the recombination expression of Gly463 mutants produced more inclusion bodies than the wild type. Consistent with the discovery that Gly463 is critical to protein conformation, mutations in such residue may also sacrifice some of the cost of protein folding. But after the mutant is correctly folded, it is stable and active. We have added this discussion to Page 13 line 7â15, method to Page 18 line 9â12.

7) P8. âThe dynamic characteristic of the pocket was also visualized using dynamic crosscorrelation maps (DCCM).â Please explain what positive and negative correlations on DCCM mean. Or, provide a reference here.

Response: We thank the reviewer for the kind suggestion. We added the following explanations and references on Page 9 line 11â19:

The dynamic characteristics of the pocket were also visualized using dynamic cross-correlation maps (DCCM)³⁷, a method that quantifies the correlation of movements between pairs of residues within a protein over the course of molecular dynamics simulations. Specifically, positive correlation values observed in DCCM indicate synchronized movements between residue pairs, moving in the same direction³⁸, which usually reflects a structurally rigid interaction or coordination within the protein structure. Conversely, negative correlation values signify that the movements of residue pairs are opposite to each other, a phenomenon that often contributes to structural flexibility and dynamic interactions within the protein.

References:

37 Ichiye, T. & Karplus, M. Collective motions in proteins: a covariance analysis of atomic fluctuations in molecular dynamics and normal mode simulations. *Proteins* 11, 205-217 (1991).

38 Arnold, G. E. & Ornstein, R. L. Molecular dynamics study of time-correlated protein domain motions and molecular flexibility: cytochrome P450BM-3. *Biophysical journal* 73, 1147-1159 (1997).

8) In Figure 4C and E, the protein sequence of ThrRS starts from 242. Why the N-terminal part of ThrRS was not analyzed?

Response: We apologize for the lack of clarity here. The N-terminal part of EcThrRS (residues 1â225) contains two domains involved in the proofreading of aminoacyl-tRNA (Supplementary Fig. 10). The constraints between this part and the catalytic domain (residues 242â535) and the anticodon-binding domains (residues 535â642) are weak. Therefore, the N-terminal part of EcThrRS was not included in our MD simulations. We have added this description to Page 9 line 20â24.

9) Both RMSD and r.m.s.d are used as abbreviations for Root Mean Square Deviation in the manuscript. Need to be unified.

Response: We have unified the abbreviation for Root Mean Square Deviation as RMSD in the revised manuscript. We thank the reviewer for pointing out this inadequacy.

Reviewer #2 (Remarks to the Author):

This manuscript addresses several timely aspects of protein structure and dynamics and their relevance for antibiotic resistance. Threonyl-tRNA synthetase (ThrRS) is one of the twenty aminoacyl-tRNA synthetases (AARS) essential for translating the genetic code. The eukaryotic and prokaryotic versions of many, if not all AARS differ sufficiently that their two superfamilies are widely studied targets for antibiotic therapeutics. Within the bacterial clades, they are also of interest because bacteria are the source for many naturally occurring antibiotics, which are thought to reflect biochemical warfare among different bacterial strains. That creates a problem for strains that synthesize different antibiotics, because they need to develop resistant homolog genes in order to survive in the presence of the antibiotics they synthesize.

ThrRS is an example. The natural product molecule Obafuorin (OB) is a potent inhibitor of bacterial (ThrRS), and species that synthesize OB must develop ThrRS variants resistant to the compound. Previous work in the Fang laboratory have previously studied the structural biology and biochemical pharmacology of OB. In this manuscript they describe extensive studies of the structures of self-resistant (*P. fluorescens*) and sensitive (*E. coli*) ThrRS molecules that implicate the protein dynamics of the reaction in OB resistance. Specifically, they show that a glycine residue in the insertion domain between Motifs 2 and 3 of OB-sensitive *E. coli* ThrRS allows an off-path conformational change that promotes formation of a covalent bond between OB and an active-site tyrosine residue

(Y462). Y462 enters the active site from what is often called the ID or insertion domain that likely represents a module acquired late in the evolution of Class II AARS and is located between Motifs 2 and 3. This domain undergoes a significant rotation toward the active site when OB is bound, leaving Motifs 2 and 3 more or less unchanged.

The authors use biophysical and biochemical analysis of various mutated residues to show the importance of the flexibility endowed by G463 in promoting the conformational change. That conformational change itself is necessary for forming a covalent bond to the OB. One key importance of these studies lies in the fact that G463 itself is outside the immediate environment provided for the binding surface of OB and thus must act cooperatively via conformational change. Moreover, thermal denaturation studies show that unless OB can form the covalent bond to Y462, its affinity is much reduced. These studies are fortified by a series of crystal structures that permit analysis of the structural consequences of the G462S and L489M mutations. The latter mutation does not materially change the affinity for OB; the former decreases the T_m by nearly two-fold. Finally, enzymatic assays confirm that ATP-dependent threonine activation is almost the same with the L489M mutant and with the G463S mutant. Thus, the G463S mutant, which prevents the conformational change, also confers OB resistance.

Altogether, this paper was a pleasure to read. It describes a robust structural and mechanistic enzymological analysis of OB resistance and localizes it to a single residue that also slows or limits the off-path conformational change that leads to strong inhibition. It thus merits publication in *Communications Biology*.

There are numerous infelicitous examples of poor English usage that should be addressed in a minor revision before publication. One example is in the caption to Figure 6, in the sentence: "OB not forms an ester bond with Tyr462". I presume the authors intended to write something along the lines of "OB does not form an ester bond with Tyr462". The authors should consult an English speaker for assistance with the entire manuscript.

Charles W. Carter, Jr

Response: We are very grateful to the respectable pioneer in the field of AARS for the deep understanding and appreciation of our research. And we apologize for the poor English usage in the previous manuscript. We have gone over the manuscript carefully and tried our best to correct the grammatical errors.

Reviewer #3 (Remarks to the Author):

Pengfei Fang et al. discuss in their manuscript how the potency of active site selective inhibitors of threonyl-tRNA synthetase is determined by the dynamics of the protein which in turn is determined by a specific glycine in the active site. This critical glycine residue (Gly463) allows the dynamic motion of threonyl-tRNA synthetase (ThrRS) either to bring binders in the correct position to allow covalent reaction (e.g. obafuorin), or to increase the interaction with the ligand so as to firmly occlude the active site as with borrelidin. Mutation of this glycine to a steric more demanding residue afforded a more rigid protein with higher thermal stability, and thus suppressed its ability to change conformation. As a result, these mutations caused resistance of ThrRS to antibiotics that are conformationally selective.

The dynamics of the active site as described here, is a bit reminiscent of the in this journal recently described conformational changes as noticed for the class Ia LeuRS (doi.org/10.1038/s42003-022-03825-8). The dynamics as highlighted in this manuscript allowed for the whole catalytic cycle, and likewise single mutants were described influencing the overall dynamics and reducing the catalytic efficiency for this particular enzyme.

In the present manuscript the requirement for the described dynamics to enable the antibacterial effects of the studied antibiotics has been clearly shown. It is unclear however, how the catalytic efficiency is influenced by the proposed mutants. This brings us to conclude some additional assays are warranted to clearly study the effects of the proposed mutations on the enzymatic activity. The additional movies are an excellent addition to illustrate the manuscript findings.

Overall, this is a very interesting paper generally well formulated, but we have following remarks and questions on the proposed text, requiring some major but feasible changes.

Response: We sincerely thank the reviewer for acknowledging our work and reminding us of its similarity to the recent LeuRS study. We have added this information and reference in the discussion section of the revised manuscript. (Page 13 line 18-20)

1. According to the authors, the mutants do not disrupt the active conformation of ThrRS. However, do these mutants affect the catalytic activity of the enzyme? No data are provided. If there is also no negative effect found on the enzymatic activity, the change of binding affinity of

OB by the ATP hydrolysis assay is more convincing.

Response: We agree with the reviewer that it is necessary to compare the ATP hydrolysis activity of the mutants themselves. We have done this experiment, and to our surprise, we found that the ATP hydrolysis activity of the Gly463 mutants was even higher than that of the wild type (Supplementary Fig. 17), suggesting that they do not disrupt the active conformation of the protein. We have added the method to Page 18 line 9â12.

2. It is better to give some definition or name for each conformational states as otherwise it is quite difficult for the reader to understand what you want to explain in the manuscript.

Response: We have adopted to this valuable suggestion. We refer to the apo conformation as "apo state", the ATP-binding conformation as "adenylation state", and the OB-binding conformation as "open state" in the revised manuscript.

3. If the serine residue in ObaO refers to the resistance for OB, does mutation of this residue to glycine restore the sensitivity of ObaO to OB?

Response: The reviewer raised a good point. We synthesized the ObaO gene and constructed the reverse mutation (S464G) of OB resistance gene ObaO. Similarly, we utilized thermal shift assay and ATP hydrolysis assay to test if ObaO_S464G lost the resistance to OB. The TSA result showed that OB increased the T_m value of ObaO_WT by 3.2 $^{\circ}\text{C}$ but increased the T_m value of ObaO_S464G by 14.1 $^{\circ}\text{C}$, suggesting that the S464G mutation creates the ability to bind OB (Supplementary Fig. 8A). Consistently, the ATP hydrolysis assay showed that OB had no inhibitory effect on ObaO_WT but inhibited the activity of ObaO_S464G with an IC_{50} value of 3.9 μM (Supplementary Fig. 8B). Therefore, mutation of the corresponding serine to glycine restores the sensitivity of ObaO to OB. We have added these results to Page 7 line 9â18. And the new Supplementary Fig. 8 is shown below.

4. In this manuscript, 36j was selected as the positive control to compare with OB when examining the potential binding affinity. However, by checking the reference paper for structure 36j (<https://doi.org/10.1021/acs.jmedchem.2c00134>), the structure shown in Figure S2 is actually compound 36k in the cited paper. This is rather confusing to the reader. Compound 36k actually shows the best antibacterial activity, while 36j has the higher affinity but presumably less uptake. Please correct the structure.

Response: We're sorry that we drew the chemical structure of 36j wrong. We have corrected this error in the new Supplementary Fig. 3A, which is shown below. We gratefully thank the reviewer for pointing this out.

5. For TSA measurements (Fig 2,3,5), the results for 36j are also a bit strange. For these four different proteins including ThrRS WT, ThrRS_G436A, ThrRS_G436S, ThrRS_G436S_Q484A, the reported T_m is 29.4, 20.8, 22.1, 27.6, which in our opinion does not make sense according to the proposed theory. In this manuscript, G436S should provide the worst result, as the serine sidechain is bulkier and provides more constraint and the overall structure should be the least flexible. But the T_m value of ThrRS_G436A is lower than for ThrRS_G436S? In addition, the double mutation has better binding affinity for 36j and OB (T_m increased comparing the double mutant with G436S); how to explain these data?

If the TSA measurements cannot fully explain the proposed theory, the real binding affinity should be measured by a biochemical or biophysical assay (preferable action).

Response: We agree with the reviewers that the TSA experiment is only a rough assessment. Because the mutations themselves can affect the thermal stability of the protein, we introduced 36j as a control. 36j is a multisite inhibitor that mimics substrate binding and has a different mechanism of action from OB, so it should be less sensitive to conformational changes than OB. Our logic is that 36j is a strongly bound compound, and if OB forms a covalent bond with ThrRS, then its effect on the thermal stability of ThrRS (T_m) should be similar to or larger than 36j, while if the OB induced T_m of a ThrRS mutant is significantly smaller than the 36j induced T_m , it should not form a covalent bond with the mutant protein.

We think that the different T_m of the mutants induced by the same compound may be due to two factors, one is the difference in binding affinity, and the other is the difference in the stability (T_m) of the mutant itself. If a protein itself has a higher T_m , the compound induced T_m value may be somewhat lower.

To prove this idea, we used surface plasmon resonance (SPR) assay to test the dissociation equilibrium constants (KD) of 36j with these ThrRS constructs. The KD values for WT, G463S, G463S_Q484A, and G463A are 7.3 nM, 8.4 nM, 18.9 nM, and 19.5 nM, respectively (Data is shown

below). Therefore, this result shows that 36j retains a relatively stable binding ability of the key ThrRS mutants involved in this study. The double mutation did not have a better binding affinity for 36j.

The reason why the double mutation showed a larger ΔT_m may be because that the mutant itself is less stable (its own T_m is low than G463S single mutant) (Data is shown below). This may be because it loses the hydrogen bond between Y462 and Q484 compared to G463S.

We also tried to analyze OB in the SPR experiment. Possibly because OB and ThrRS_WT are covalently bound and the protein undergoes significant conformational changes, we were unable to obtain SPR data that could be analyzed.

Together, we have updated the description of the reason to use 36j as a control on Page 5 line 10-14, and added the new SPR results to the new Supplementary Fig. 3B-E.

6. Figure 2D: the electron map shown does not really correspond with the residues? Suggestion to the authors: please reconstruct the density map, the selected residues can be shown with the electron density while leaving other residues shown in sticks. This will help clarify the content.

Response: We appreciate the reviewer's suggestion very much. We have re-made Figure 2D according to this suggestion.

7. Figure 3 - the title: "Interaction of introduced Ser463 and conserved Gln484 doesn't prevent the covalent binding of OB" is confusing. We believe the presence of Ser does prevent covalent binding, (no reaction with Tyr462) but not the less deeper binding of OB. What the authors most probably mean is that the interaction between Ser463 and Gln484 is not the reason for lack of covalent binding of OB. The title can be changed to "Spatial interaction between Ser463 and Gln484 does not confer OB resistance".

Response: The reviewer is right. We have changed the title of Figure 3 to "Spatial interaction between Ser463 and Gln484 does not confer OB resistance" in the revised manuscript.

8. Figure 3 Panel D: once more, electron map and structure model do not seem to correspond?? We suggest to redo the figure as for Figure 2D and only show the density of selected residues. In addition, in this state, the OB cannot form a covalent bond with Tyr462, and hence no density for OB can be seen, but for clarity such statement should be added here to avoid confusion.

Response: We have re-made Figure 3D according to this suggestion. And we have added the description that "In this state, the OB cannot form a covalent bond with Tyr462; hence, no density for OB can be seen."

9. Figure 6: Phe378 in figure 6C needs to be corrected to Phe379. In addition, the stacking of nitrobenzyl between Arg520 and Phe379 can be (difficultly) seen, this group is superposed with adenine in this view. Hence, preferably add to the text "The nitrophenyl group of OB (in grey) binds to". For clarity, also include in text that the orange parts are both terminal phosphate moieties of the co-crystallized ATP.

Response: We have added a transparent white background to the Phe379 to make it more visible. And we have also modified the legend according to the reviewer's suggestion. It now reads "The nitrophenyl group of OB (in gray) binds between Phe379 and Arg520 where the adenine group of ATP (in purple blue) stacks. The orange parts in the middle of the panel are both terminal phosphate moieties of the co-crystallized ATP."

10. Please use the required format to provide all references. (e.g. the title format of references is not consistent).

Response: We have updated the format of the reference to make it conform to the requirement.

Additional detailed remarks:

Page 6 line 3: please add the reference for the assay used.

Response: We have added the reference in the revised manuscript Page 5 line 9.

Page 7 line 10: to clarify, add: while in the OB-bound structure of the wild type

Response: We have added "of the wild type" after the text "while in the OB-bound structure" (Page 8 line 4 in the revised manuscript).

Page 9 halfway: we believe this can be better phrased as "so that it can be suppressed by OB when DR-A reaches 17 Å."

Response: We have rephrased this sentence as "This finding suggests that the pocket of EcThrRS_G463A cannot be extended so that its DR-A reaches 17 Å and can be effectively suppressed by OB." (Page 10 line 23 in the revised manuscript).

Page 9 penultimate line: "nadirs"?

Response: We have corrected the improper plural form and the repetition of the meaning of "nadir" and "at the lowest energy". Now this sentence reads "By comparing the conformation of residues 419-467 in the FEL nadir structure of the three proteins, we found that EcThrRS_WT had the most extended conformation, followed by G463A, and the narrowest was G463S." (new Page 11 line 4-7)

Page 10 line 2: "WT is closer to the OB-bound form". We are a bit lost here. Is the WT structure not the same as for the apo protein? Or did the authors mean the WT in presence of ATP?

Response: We apologize for the lack of clarity here. We were comparing the difference between the energy nadir conformation in the computational simulation and the real crystal structures. We have rephrased this sentence as "by comparing the FEL nadir conformation in the computational simulation and the real crystal structures, we found that the FEL nadir conformation of EcThrRS_WT was closer to the open state crystal structure, while the FEL nadir conformations of EcThrRS_G463S and EcThrRS_G463A were more similar to the apo state crystal structures." (Page 11 line 7-11)

Page 12 line 5: "MD is also thought to influence drug resistance". This seems not correctly formulated; MD itself not influences drug resistance, but rather gives insight into the reasons for potential resistance development.

Response: We have rephrased the text as suggested by the reviewer: "MD is also thought to give insight into the reasons for potential resistance development" (Page 14 line 2-3 in the revised manuscript).

Page 12 penultimate line: "OB-producing *P. fluorescens*". Maybe add in parenthesis for clarity: "(equipped with the G463S mutation in the ObaO protein)".

Response: We have rephrased this as "OB-producing *P. fluorescens* that is equipped with a Serine substitution at residue 464 (corresponding to Gly463 in EcThrRS) in the ObaO protein." (Page 14 line 11-13)

Page 13: the key resources table is obviously important but it is rather awkward to have a 3 page table in the main section. This could alternatively be in the supplementary section. I leave this remark to the discretion of the editor.

Response: We have moved the key resources table to the supplementary section P28-31.

Page 15 line 5 from bottom: typo; correct to "washed".

Response: We have corrected this typo on new Page 15 line 9.

Figure 1D: for convenience, use an arrow to pinpoint either the Tyr462 or Gly463 residue.

Response: We have added arrows to pinpoint the Tyr462 residue in this Figure panel and sequence alignment Figure 1C and Figure S2.

Figure 5: correct the second part of the title to "blockage on the covalent binding of OB which were observed in the mutant G463S."

Response: We have corrected the second part of the title to "blockage on the covalent binding of OB which were observed in the mutant G463S."

Figure 5 panel E: please indicate some residues on the cartoon, at least position 463.

Response: We have indicated positions 462 and 463 in the revised Figure.

Figure 7 title: should read "resistance of EcThrRS to BN"

Response: We have changed the title to "Gly463 mutations also confer resistance of EcThrRS to BN".

Table S2: please add the PDB code in the table.

Response: We have added the PDB code "8WIJ" of EcThrRS_L489M OB structure to Table S2.

Version 1:

Reviewer comments:

Reviewer #1

(Remarks to the Author)

The authors have perfectly addressed my concerns. I recommend this revised version to be published.

Reviewer #2

(Remarks to the Author)

The authors have responded in depth to the concerns of all three reviewers. The manuscript merits publication in Nature Communications Biology.

Reviewer #3

(Remarks to the Author)

The manuscript of Pengfei Fang et al. has been considerably improved, answering all questions and remarks of the various reviewers. A thorough rebuttal highlighting all changes was provided along with the revised paper. We did not find further flaws or uncertainties and believe this is an outstanding piece of work deserving publication. Although maybe unusual we would like to congratulate the authors with their work!

Author Rebuttal letter:

RE: manuscript COMMSBIO-24-1169A by Qiao et al.

All three reviewers have approved our revised manuscript. We would like to thank the reviewers again for their constructive comments.

REVIEWERS' COMMENTS:

Reviewer #1 (Remarks to the Author):

The authors have perfectly addressed my concerns. I recommend this revised version to be published.

Reviewer #2 (Remarks to the Author):

The authors have responded in depth to the concerns of all three reviewers. The manuscript merits publication in Nature Communications Biology.

Reviewer #3 (Remarks to the Author):

The manuscript of Pengfei Fang et al. has been considerably improved, answering all questions and remarks of the various reviewers. A thorough rebuttal highlighting all changes was provided along with the revised paper. We did not find further flaws or uncertainties and believe this is an outstanding piece of work deserving publication. Although maybe unusual we would like to congratulate the authors with their work!
